# Gradient Estimation with Stochastic Softmax Tricks

**Max B. Paulus**[*]
ETH Zürich
max.paulus@inf.ethz.ch

**Dami Choi**[*]
University of Toronto
choidami@cs.toronto.edu

**Daniel Tarlow**
Google Research, Brain Team
dtarlow@google.com

**Andreas Krause**
ETH Zürich
krausea@ethz.ch

**Chris J. Maddison**[†]
University of Toronto & DeepMind
cmaddis@cs.toronto.edu

## Abstract

The Gumbel-Max trick is the basis of many relaxed gradient estimators. These estimators are easy to implement and low variance, but the goal of scaling them comprehensively to large combinatorial distributions is still outstanding. Working within the perturbation model framework, we introduce stochastic softmax tricks, which generalize the Gumbel-Softmax trick to combinatorial spaces. Our framework is a unified perspective on existing relaxed estimators for perturbation models, and it contains many novel relaxations. We design structured relaxations for subset selection, spanning trees, arborescences, and others. When compared to less structured baselines, we find that stochastic softmax tricks can be used to train latent variable models that perform better and discover more latent structure.

## 1 Introduction

Gradient computation is the methodological backbone of deep learning, but computing gradients is not always easy. Gradients with respect to parameters of the density of an integral are generally intractable, and one must resort to gradient estimators [8, 61]. Typical examples of objectives over densities are returns in reinforcement learning [76] or variational objectives for latent variable models [e.g., 37, 68]. In this paper, we address *gradient estimation for discrete distributions* with an emphasis on latent variable models. We introduce a relaxed gradient estimation framework for combinatorial discrete distributions that generalizes the Gumbel-Softmax and related estimators [53, 35].

Relaxed gradient estimators incorporate bias in order to reduce variance. Most relaxed estimators are based on the Gumbel-Max trick [52, 54], which reparameterizes distributions over one-hot binary vectors. The Gumbel-Softmax estimator is the simplest; it continuously approximates the Gumbel-Max trick to admit a reparameterization gradient [37, 68, 72]. This is used to optimize the "soft" approximation of the loss as a surrogate for the "hard" discrete objective.

Adding structured latent variables to deep learning models is a promising direction for addressing a number of challenges: improving interpretability (e.g., via latent variables for subset selection [17] or parse trees [19]), incorporating problem-specific constraints (e.g., via enforcing alignments [58]), and improving generalization (e.g., by modeling known algorithmic structure [30]). Unfortunately, the vanilla Gumbel-Softmax cannot scale to distributions over large state spaces, and the development of structured relaxations has been piecemeal.

We introduce *stochastic softmax tricks* (SSTs), which are a unified framework for designing structured relaxations of combinatorial distributions. They include relaxations for the above applications, as well

---

[*]Equal Contribution. Correspondence to max.paulus@inf.ethz.ch, choidami@cs.toronto.edu.
[†]Work done partly at the Institute for Advanced Study, Princeton, NJ.

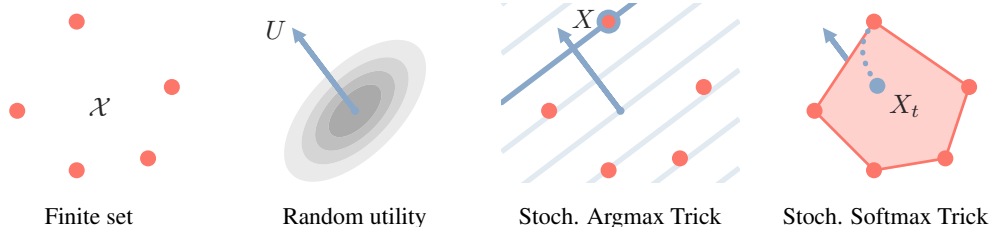

<center>Finite set      Random utility      Stoch. Argmax Trick      Stoch. Softmax Trick</center>

Figure 1: Stochastic softmax tricks relax discrete distributions that can be reparameterized as random linear programs. $X$ is the solution of a random linear program defined by a finite set $\mathcal{X}$ and a random utility $U$ with parameters $\theta \in \mathbb{R}^m$. To design relaxed gradient estimators with respect to $\theta$, $X_t$ is the solution of a random convex program that continuously approximates $X$ from within the convex hull of $\mathcal{X}$. The Gumbel-Softmax [53, 35] is an example of a stochastic softmax trick.

as many novel ones. To use an SST, a modeler chooses from a class of models that we call *stochastic argmax tricks* (SMT). These are instances of perturbation models [e.g., 64, 33, 78, 27], and they induce a distribution over a finite set $\mathcal{X}$ by optimizing a linear objective (defined by random utility $U \in \mathbb{R}^n$) over $\mathcal{X}$. An SST relaxes this SMT by combining a strongly convex regularizer with the random linear objective. The regularizer makes the solution a continuous, a.e. differentiable function of $U$ and appropriate for estimating gradients with respect to $U$'s parameters. The Gumbel-Softmax is a special case. Fig. 1 provides a summary.

We test our relaxations in the Neural Relational Inference (NRI) [38] and L2X [17] frameworks. Both NRI and L2X use variational losses over latent combinatorial distributions. When the latent structure in the model matches the true latent structure, we find that our relaxations encourage the unsupervised discovery of this combinatorial structure. This leads to models that are more interpretable and achieve stronger performance than less structured baselines. All proofs are in the Appendix.

## 2    Problem Statement

Let $\mathcal{Y}$ be a non-empty, finite set of combinatorial objects, e.g. the spanning trees of a graph. To represent $\mathcal{Y}$, define the embeddings $\mathcal{X} \subseteq \mathbb{R}^n$ of $\mathcal{Y}$ to be the image $\{\mathrm{rep}(y) \mid y \in \mathcal{Y}\}$ of some embedding function $\mathrm{rep} : \mathcal{Y} \to \mathbb{R}^n$.[3] For example, if $\mathcal{Y}$ is the set of spanning trees of a graph with edges $E$, then we could enumerate $y_1, \ldots, y_{|\mathcal{Y}|}$ in $\mathcal{Y}$ and let $\mathrm{rep}(y)$ be the one-hot binary vector of length $|\mathcal{Y}|$, with $\mathrm{rep}(y)_i = 1$ iff $y = y_i$. This requires a very large ambient dimension $n = |\mathcal{Y}|$. Alternatively, in this case we could use a more efficient, structured representation: $\mathrm{rep}(y)$ could be a binary indicator vector of length $|E| \ll |\mathcal{Y}|$, with $\mathrm{rep}(y)_e = 1$ iff edge $e$ is in the tree $y$. See Fig. 2 for visualizations and additional examples of structured binary representations. We assume that $\mathcal{X}$ is convex independent.[4]

Given a probability mass function $p_\theta : \mathcal{X} \to (0, 1]$ that is differentiable in $\theta \in \mathbb{R}^m$, a loss function $\mathcal{L} : \mathbb{R}^n \to \mathbb{R}$, and $X \sim p_\theta$, our ultimate goal is gradient-based optimization of $\mathbb{E}[\mathcal{L}(X)]$. Thus, we are concerned in this paper with the problem of estimating the derivatives of the expected loss,

$$\frac{d}{d\theta}\mathbb{E}[\mathcal{L}(X)] = \frac{d}{d\theta}\left(\sum_{x \in \mathcal{X}} \mathcal{L}(x) p_\theta(x)\right). \tag{1}$$

## 3    Background on Gradient Estimation

Relaxed gradient estimators assume that $\mathcal{L}$ is differentiable and use a change of variables to remove the dependence of $p_\theta$ on $\theta$, known as the reparameterization trick [37, 68]. The Gumbel-Softmax trick (GST) [53, 35] is a simple relaxed gradient estimator for one-hot embeddings, which is based on the Gumbel-Max trick (GMT) [52, 54]. Let $\mathcal{X}$ be the one-hot embeddings of $\mathcal{Y}$ and $p_\theta(x) \propto \exp(x^T \theta)$.

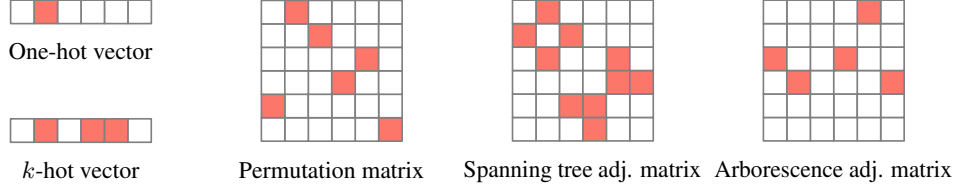

One-hot vector

$k$-hot vector     Permutation matrix     Spanning tree adj. matrix     Arborescence adj. matrix

Figure 2: Structured discrete objects can be represented by binary arrays. In these graphical representations, color indicates 1 and no color indicates 0. For example, "Spanning tree" is the adjacency matrix of an undirected spanning tree over 6 nodes; "Arborescence" is the adjacency matrix of a directed spanning tree rooted at node 3.

The GMT is the following identity: for $X \sim p_\theta$ and $G_i + \theta_i \sim \text{Gumbel}(\theta_i)$ indep.,

$$X \overset{d}{=} \arg\max_{x \in \mathcal{X}} (G + \theta)^T x. \tag{2}$$

Ideally, one would have a reparameterization estimator, $\mathbb{E}[d\mathcal{L}(X)/d\theta] = d\,\mathbb{E}[\mathcal{L}(X)]/d\theta$,[5] using the right-hand expression in (2). Unfortunately, this fails. The problem is not the lack of differentiability, as normally reported. In fact, the argmax is differentiable almost everywhere. Instead it is the jump discontinuities in the argmax that invalidate this particular exchange of expectation and differentiation [48, 8, Chap. 7.2]. The GST estimator [53, 35] overcomes this by using the tempered softmax, $\text{softmax}_t(u)_i = \exp(u_i/t)/\sum_{j=1}^n \exp(u_j/t)$ for $u \in \mathbb{R}^n, t > 0$, to continuously approximate $X$,

$$X_t = \text{softmax}_t(G + \theta). \tag{3}$$

The relaxed estimator is $d\mathcal{L}(X_t)/d\theta$. While this is a biased estimator of (1), it is an unbiased estimator of $d\mathbb{E}[\mathcal{L}(X_t)]/d\theta$ and $X_t \to X$ a.s. as $t \to 0$. Thus, $d\mathcal{L}(X_t)/d\theta$ is used for optimizing $\mathbb{E}[\mathcal{L}(X_t)]$ as a surrogate for $\mathbb{E}[\mathcal{L}(X)]$, on which the final model is evaluated.

The score function estimator [28, 84], $\mathcal{L}(X)\,\partial \log p_\theta(X)/\partial\theta$, is the classical alternative. It is a simple, unbiased estimator, but without highly engineered control variates, it suffers from high variance [60]. Building on the score function estimator are a variety of estimators that require multiple evaluations of $\mathcal{L}$ to reduce variance [32, 81, 29, 87, 45, 9]. The advantages of relaxed estimators are the following: they only require a single evaluation of $\mathcal{L}$, they are easy to implement using modern software packages [1, 65, 16], and, as reparameterization gradients, they tend to have low variance [26].

## 4    Stochastic Argmax Tricks

Simulating a GST requires enumerating $|\mathcal{Y}|$ random variables, so it cannot scale. We overcome this by identifying generalizations of the GMT that can be relaxed and that scale to large $\mathcal{Y}$s by exploiting structured embeddings $\mathcal{X}$. We call these *stochastic argmax tricks* (SMTs), because they are perturbation models [78, 27], which can be relaxed into stochastic softmax tricks (Section 5).

**Definition 1.** *Given a non-empty, convex independent, finite set $\mathcal{X} \subseteq \mathbb{R}^n$ and a random utility $U$ whose distribution is parameterized by $\theta \in \mathbb{R}^m$, a* stochastic argmax trick *for $X$ is the linear program,*

$$X = \arg\max_{x \in \mathcal{X}} U^T x. \tag{4}$$

The GMT is recovered with one-hot $\mathcal{X}$ and $U \sim \text{Gumbel}(\theta)$. We assume that (4) is a.s. unique, which is guaranteed if $U$ a.s. never lands in any particular lower dimensional subspace (Prop. 3, App. A). Because efficient linear solvers are known for many structured $\mathcal{X}$, SMTs are capable of scaling to very large $\mathcal{Y}$ [74, 41, 40]. For example, if $\mathcal{X}$ are the edge indicator vectors of spanning trees $\mathcal{Y}$, then (4) is the maximum spanning tree problem, which is solved by Kruskal's algorithm [46].

The role of the SMT in our framework is to reparameterize $p_\theta$ in (1). Ideally, *given $p_\theta$*, there would be an efficient (e.g., $\mathcal{O}(n)$) method for simulating *some $U$* such that the marginal of $X$ in (4) is $p_\theta$. The GMT shows that this is possible for one-hot $\mathcal{X}$, but the situation is not so simple for structured

$\mathcal{X}$. Characterizing the marginal of $X$ in general is difficult [78, 34], but $U$ that are efficient to sample from typically induce conditional independencies in $p_\theta$ [27]. Therefore, we are not able to reparameterize an arbitrary $p_\theta$ on structured $\mathcal{X}$. Instead, for structured $\mathcal{X}$ we *assume* that $p_\theta$ is reparameterized by (4), and treat $U$ as a modeling choice. Thus, we caution against the standard approach of taking $U \sim \mathrm{Gumbel}(\theta)$ or $U \sim \mathcal{N}(\theta, \sigma^2 I)$ without further analysis. Practically, in experiments we show that the difference in noise distribution can have a large impact on quantitative results. Theoretically, we show in App. B that an SMT over directed spanning trees with negative exponential utilities has a more interpretable structure than the same SMT with Gumbel utilities.

## 5 Stochastic Softmax Tricks

If we assume that $X \sim p_\theta$ is reparameterized as an SMT, then a stochastic softmax trick (SST) is a random convex program with a solution that relaxes $X$. An SST has a valid reparameterization gradient estimator. Thus, we propose using SSTs as surrogates for estimating gradients of (1), a generalization of the Gumbel-Softmax approach. Because we want gradients with respect to $\theta$, we assume that $U$ is also reparameterizable.

Given an SMT, an SST incorporates a strongly convex regularizer to the linear objective, and expands the state space to the convex hull of the embeddings $\mathcal{X} = \{x_1, \dots, x_m\} \subseteq \mathbb{R}^n$,

$$P := \mathrm{conv}(\mathcal{X}) := \left\{ \sum_{i=1}^m \lambda_i x_i \,\middle|\, \lambda_i \geq 0, \ \sum_{i=1}^m \lambda_i = 1 \right\}. \tag{5}$$

Expanding the state space to a convex polytope makes it path-connected, and the strongly convex regularizer ensures that the solutions are continuous over the polytope.

**Definition 2.** *Given a stochastic argmax trick* $(\mathcal{X}, U)$ *where* $P := \mathrm{conv}(\mathcal{X})$ *and a proper, closed, strongly convex function* $f : \mathbb{R}^n \to \{\mathbb{R}, \infty\}$ *whose domain contains the relative interior of* $P$, *a* stochastic softmax trick *for* $X$ *at temperature* $t > 0$ *is the convex program,*

$$X_t = \arg\max_{x \in P} U^T x - t f(x) \tag{6}$$

For one-hot $\mathcal{X}$, the Gumbel-Softmax is a special case of an SST where $P$ is the probability simplex, $U \sim \mathrm{Gumbel}(\theta)$, and $f(x) = \sum_i x_i \log(x_i)$. Objectives like (6) have a long history in convex analysis [e.g., 69, Chap. 12] and machine learning [e.g., 83, Chap. 3]. In general, the difficulty of computing the SST will depend on the interaction between $f$ and $\mathcal{X}$.

$X_t$ is suitable as an approximation of $X$. At positive temperatures $t$, $X_t$ is a function of $U$ that ranges over the faces and relative interior of $P$. The degree of approximation is controlled by the temperature parameter, and as $t \to 0^+$, $X_t$ is driven to $X$ a.s.

**Proposition 1.** *If* $X$ *in Def.* 1 *is a.s. unique, then for* $X_t$ *in Def.* 2, $\lim_{t \to 0^+} X_t = X$ *a.s. If additionally* $\mathcal{L} : P \to \mathbb{R}$ *is bounded and continuous, then* $\lim_{t \to 0^+} \mathbb{E}[\mathcal{L}(X_t)] = \mathbb{E}[\mathcal{L}(X)]$.

It is common to consider temperature parameters that interpolate between marginal inference and a deterministic, most probable state. While superficially similar, our relaxation framework is different; as $t \to 0^+$, an SST approaches *a sample from the SMT model* as opposed to a deterministic state.

$X_t$ also admits a reparameterization trick. The SST reparameterization gradient estimator given by,

$$\frac{d\mathcal{L}(X_t)}{d\theta} = \frac{\partial \mathcal{L}(X_t)}{\partial X_t} \frac{\partial X_t}{\partial U} \frac{dU}{d\theta}. \tag{7}$$

If $\mathcal{L}$ is differentiable on $P$, then this is an unbiased estimator[6] of the gradient $d\mathbb{E}[\mathcal{L}(X_t)]/d\theta$, because $X_t$ is continuous and a.e. differentiable:

**Proposition 2.** $X_t$ *in Def.* 2 *exists, is unique, and is a.e. differentiable and continuous in* $U$.

In general, the Jacobian $\partial X_t / \partial U$ will need to be derived separately given a choice of $f$ and $\mathcal{X}$. However, as pointed out by [21], because the Jacobian of $X_t$ symmetric [70, Cor. 2.9], local finite difference approximations can be used to approximate $d\mathcal{L}(X_t)/dU$ (App. D). These finite difference approximations only require two additional calls to a solver for (6) and do not require additional evaluations of $\mathcal{L}$. We found them to be helpful in a few experiments (c.f., Section 8).

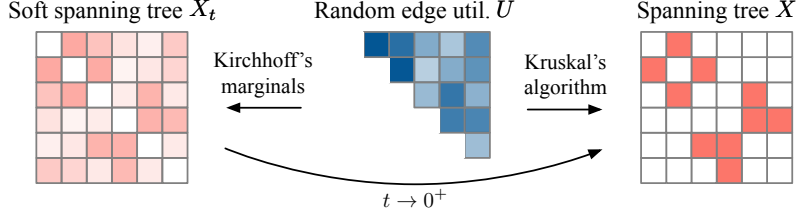

Figure 3: An example realization of a spanning tree SST for an undirected graph. Middle: Random undirected edge utilities. Left: The random soft spanning tree $X_t$, represented as a weighted adjacency matrix, can be computed via Kirchhoff's Matrix-Tree theorem. Right: The random spanning tree $X$, represented as an adjacency matrix, can be computed with Kruskal's algorithm.

There are many, well-studied $f$ for which (6) is efficiently solvable. If $f(x) = \|x\|^2/2$, then $X_t$ is the Euclidean projection of $U/t$ onto $P$. Efficient projection algorithms exist for some convex sets [see 85, 23, 50, 13, and references therein], and more generic algorithms exist that only call linear solvers as subroutines [63]. In some of the settings we consider, generic negative-entropy-based relaxations are also applicable. We refer to relaxations with $f(x) = \sum_{i=1}^n x_i \log(x_i)$ as *categorical entropy relaxations* [e.g., 13, 14]. We refer to relaxations with $f(x) = \sum_{i=1}^n x_i \log(x_i) + (1-x_i)\log(1-x_i)$ as *binary entropy relaxations* [e.g., 7].

Marginal inference in exponential families is a rich source of SST relaxations. Consider an exponential family over the finite set $\mathcal{X}$ with natural parameters $u/t \in \mathbb{R}^n$ such that the probability of $x \in \mathcal{X}$ is proportional to $\exp(u^T x/t)$. The *marginals* $\mu_t : \mathbb{R}^n \to \mathrm{conv}(\mathcal{X})$ of this family are solutions of a convex program in exactly the form (6) [83], i.e., there exists $A^* : \mathrm{conv}(\mathcal{X}) \to \{\mathbb{R}, \infty\}$ such that,

$$\mu_t(u) := \sum_{x \in \mathcal{X}} \frac{x \exp(u^T x/t)}{\sum_{y \in \mathcal{X}} \exp(u^T y/t)} = \arg\max_{x \in P} u^T x - t A^*(x). \qquad (8)$$

The definition of $A^*$, which generates $\mu_t$ in (8), can be found in [83, Thm. 3.4]. $A^*$ is a kind of negative entropy and in our case it satisfies the assumptions in Def. 2. Computing $\mu_t$ amounts to marginal inference in the exponential family, and efficient algorithms are known in many cases [see 83, 40], including those we consider. We call $X_t = \mu_t(U)$ the *exponential family entropy relaxation*.

Taken together, Prop. 1 and 2 suggest our proposed use for SSTs: optimize $\mathbb{E}[\mathcal{L}(X_t)]$ at a positive temperature, where unbiased gradient estimation is available, but evaluate $\mathbb{E}[\mathcal{L}(X)]$. We find that this works well in practice if the temperature used during optimization is treated as a hyperparameter and selected over a validation set. It is worth emphasizing that the choice of relaxation is unrelated to the distribution $p_\theta$ of $X$ in the corresponding SMT. $f$ is not only a modeling choice; it is a computational choice that will affect the cost of computing (6) and the quality of the gradient estimator.

## 6 Examples of Stochastic Softmax Tricks

The Gumbel-Softmax [53, 35] introduced neither the Gumbel-Max trick nor the softmax. The novelty of this work is neither the pertubation model framework nor the relaxation framework in isolation, but their combined use for gradient estimation. Here we layout some example SSTs, organized by the set $\mathcal{Y}$ with a choice of embeddings $\mathcal{X}$. Bold italics indicates previously described relaxations, most of which are bespoke and not describable in our framework. Italics indicates our novel SSTs used in our experiments; some of these are also novel perturbation models. A complete discussion is in App. B.

**Subset selection.** $\mathcal{X}$ is the set of binary vectors indicating membership in the subsets of a finite set $S$. *Indep. S* uses $U \sim \mathrm{Logistic}(\theta)$ and a binary entropy relaxation. $X$ and $X_t$ are computed with a dimension-wise step function or sigmoid, resp.

**k-Subset selection.** $\mathcal{X}$ is the set of binary vectors with a $k$-hot binary vectors indicating membership in a $k$-subset of a finite set $S$. All of the following SMTs use $U \sim \mathrm{Gumbel}(\theta)$. Our SSTs use the following relaxations: euclidean [6] and categorical [56], binary [7], and exponential family [77] entropies. $X$ is computed by sorting $U$ and setting the top $k$ elements to 1 [13]. *R Top k* refers to our SST with relaxation $R$. **L2X** [17] and **SoftSub** [86] are bespoke relaxations.

**Correlated k-subset selection.** $\mathcal{X}$ is the set of $(2n-1)$-dimensional binary vectors with a $k$-hot cardinality constraint on the first $n$ dimensions and a constraint that the $n-1$ dimensions indicate correlations between adjacent dimensions in the first $n$, i.e. the vertices of the correlation polytope of a chain [83, Ex. 3.8] with an added cardinality constraint [59]. *Corr. Top $k$* uses $U_{1:n} \sim \text{Gumbel}(\theta_{1:n})$, $U_{n+1:2n-1} = \theta_{n+1:2n-1}$, and the exponential family entropy relaxation. $X$ and $X_t$ can be computed with dynamic programs [79], see App. B.

**Perfect Bipartite Matchings.** $\mathcal{X}$ is the set of $n \times n$ permutation matrices representing the perfect matchings of the complete bipartite graph $K_{n,n}$. The ***Gumbel-Sinkhorn*** [58] uses $U \sim \text{Gumbel}(\theta)$ and a Shannon entropy relaxation. $X$ can be computed with the Hungarian method [47] and $X_t$ with the Sinkhorn algorithm [75]. ***Stochastic NeuralSort*** [31] uses correlated Gumbel-based utilities that induce a Plackett-Luce model and a bespoke relaxation.

**Undirected spanning trees.** Given a graph $(V, E)$, $\mathcal{X}$ is the set of binary indicator vectors of the edge sets $T \subseteq E$ of undirected spanning trees. *Spanning Tree* uses $U \sim \text{Gumbel}(\theta)$ and the exponential family entropy relaxation. $X$ can be computed with Kruskal's algorithm [46], $X_t$ with Kirchhoff's matrix-tree theorem [42, Sec. 3.3], and both are represented as adjacency matrices, Fig. 3.

**Rooted directed spanning trees.** Given a graph $(V, E)$, $\mathcal{X}$ is the set of binary indicator vectors of the edge sets $T \subseteq E$ of $r$-rooted, directed spanning trees. *Arborescence* uses $U \sim \text{Gumbel}(\theta)$ or $-U \sim \text{Exp}(\theta)$ or $U \sim \mathcal{N}(\theta, I)$ and an exponential family entropy relaxation. $X$ can be computed with the Chu-Liu-Edmonds algorithm [18, 24], $X_t$ with a directed version of Kirchhoff's matrix-tree theorem [42, Sec. 3.3], and both are represented as adjacency matrices. ***Perturb & Parse*** [19] further restricts $\mathcal{X}$ to be projective trees, uses $U \sim \text{Gumbel}(\theta)$, and uses a bespoke relaxation.

## 7 Related Work

Here we review perturbation models (PMs) and methods for relaxation more generally. SMTs are a subclass of PMs, which draw samples by optimizing a random objective. Perhaps the earliest example comes from Thurstonian ranking models [80], where a distribution over rankings is formed by sorting a vector of noisy scores. Perturb & MAP models [64, 33] were designed to approximate the Gibbs distribution over a combinatorial output space using low-order, additive Gumbel noise. Randomized Optimum models [78, 27] are the most general class, which include non-additive noise distributions and non-linear objectives. Recent work [51] uses PMs to construct finite difference approximations of the expected loss' gradient. It requires optimizing a non-linear objective over $\mathcal{X}$, and making this applicable to our settings would require significant innovation.

Using SSTs for gradient estimation requires differentiating through a convex program. This idea is not ours and is enjoying renewed interest in [3, 4, 5]. In addition, specialized solutions have been proposed for quadratic programs [6, 55, 15] and linear programs with entropic regularizers over various domains [56, 7, 2, 58, 15]. In graphical modeling, several works have explored differentiating through marginal inference [21, 71, 67, 22, 77, 20] and our exponential family entropy relaxation builds on this work. The most superficially similar work is [11], which uses noisy utilities to smooth the solutions of linear programs. In [11], the noise is a tool for approximately relaxing a deterministic linear program. Our framework uses relaxations to approximate *stochastic* linear programs.

## 8 Experiments

Our goal in these experiments was to evaluate the use of SSTs for learning distributions over structured latent spaces in deep structured models. We chose frameworks (NRI [38], L2X [17], and a latent parse tree task) in which relaxed gradient estimators are the methods of choice, and investigated the effects of $\mathcal{X}$, $f$, and $U$ on the task objective and on the unsupervised structure discovery. For NRI, we also implemented the standard single-loss-evaluation score function estimators (REINFORCE [84] and NVIL [60]), and the best SST outperformed these baselines both in terms of average performance and variance, see App. C. All SST models were trained with the "soft" SST and evaluated with the "hard" SMT. We optimized hyperparameters (including fixed training temperature $t$) using random search over multiple independent runs. We selected models on a validation set according to the best objective value obtained during training. All reported values are measured on a test set. Error bars are bootstrap standard errors over the model selection process. We refer to SSTs defined in Section 6 with italics. Details are in App. D. Code is available at `https://github.com/choidami/sst`.

Table 1 & Figure 4: *Spanning Tree* performs best on structure recovery, despite being trained on the ELBO. Test ELBO and structure recovery metrics are shown from models selected on valid. ELBO. Below: Test set example where *Spanning Tree* recovers the ground truth latent graph perfectly.

| | $T = 10$ | | | $T = 20$ | | |
|---|---|---|---|---|---|---|
| Edge Distribution | ELBO | Edge Prec. | Edge Rec. | ELBO | Edge Prec. | Edge Rec. |
| *Indep. Directed Edges* [38] | $-1370 \pm 20$ | $48 \pm 2$ | $\mathbf{93 \pm 1}$ | $-1340 \pm 160$ | $97 \pm 3$ | $99 \pm 1$ |
| *E.F. Ent. Top* $\lvert V \rvert - 1$ | $-2100 \pm 20$ | $41 \pm 1$ | $41 \pm 1$ | $-1700 \pm 320$ | $98 \pm 6$ | $98 \pm 6$ |
| *Spanning Tree* | $\mathbf{-1080 \pm 110}$ | $\mathbf{91 \pm 3}$ | $91 \pm 3$ | $\mathbf{-1280 \pm 10}$ | $\mathbf{99 \pm 1}$ | $\mathbf{99 \pm 1}$ |

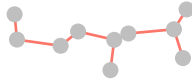 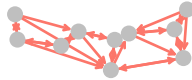 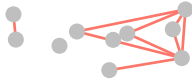 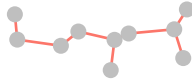

Ground Truth        *Indep. Directed Edges*        *E.F. Ent. Top* $\lvert V \rvert - 1$        *Spanning Tree*

## 8.1 Neural Relational Inference (NRI) for Graph Layout

With NRI we investigated the use of SSTs for latent structure recovery and final performance. NRI is a graph neural network (GNN) model that samples a latent interaction graph $G = (V, E)$ and runs messages over the adjacency matrix to produce a distribution over an interacting particle system. NRI is trained as a variational autoencoder to maximize a lower bound (ELBO) on the marginal log-likelihood of the time series. We experimented with three SSTs for the encoder distribution: *Indep. Binary* over directed edges, which is the baseline NRI encoder [38], *E.F. Ent. Top* $\lvert V \rvert - 1$ over undirected edges, and *Spanning Tree* over undirected edges. We computed the KL with respect to the random utility $U$ for all SSTs; see App. D for details. Our dataset consisted of latent prior spanning trees over 10 vertices sampled from the $\mathrm{Gumbel}(0)$ prior. Given a tree, we embed the vertices in $\mathbb{R}^2$ by applying $T \in \{10, 20\}$ iterations of a force-directed algorithm [25]. The model saw particle locations at each iteration, not the underlying spanning tree.

We found that *Spanning Tree* performed best, improving on both ELBO and the recovery of latent structure over the baseline [38]. For structure recovery, we measured edge precision and recall against the ground truth adjacency matrix. It recovered the edge structure well even when given only a short series ($T = 10$, Fig. 4). Less structured baselines were only competitive on longer time series.

## 8.2 Unsupervised Parsing on ListOps

We investigated the effect of $\mathcal{X}$'s structure and of the utility distribution in a latent parse tree task. We used a simplified variant of the ListOps dataset [62], which contains sequences of prefix arithmetic expressions, e.g., `max[ 3 min[ 8 2 ]]`, that evaluate to an integer in $[0, 9]$. The arithmetic syntax induces a directed spanning tree rooted at its first token with directed edges from operators to operands. We modified the data by removing the `summod` operator, capping the maximum depth of the ground truth dependency parse, and capping the maximum length of a sequence. This simplifies the task considerably, but it makes the problem accessible to GNN models of fixed depth. Our models used a bi-LSTM encoder to produce a distribution over edges (directed or undirected) between all pairs of tokens, which induced a latent (di)graph. Predictions were made from the final embedding of the first token after passing messages in a GNN architecture over the latent graph. For undirected graphs, messages were passed in both directions. We experimented with the following SSTs for the edge distribution: *Indep. Undirected Edges*, *Spanning Tree*, *Indep. Directed Edges*, and *Arborescence* (with three separate utility distributions). *Arborescence* was rooted at the first token. For baselines we used an unstructured LSTM and the GNN over the ground truth parse. All models were trained with cross-entropy to predict the integer evaluation of the sequence.

The best performing models were structured models whose structure better matched the true latent structure (Table 2). For each model, we measured the accuracy of its prediction (task accuracy). We measured both precision and recall with respect to the ground truth parse's adjacency matrix. [7] Both tree-structured SSTs outperformed their independent edge counterparts on all metrics. Overall,

Table 2: Matching ground truth structure (non-tree → tree) improves performance on ListOps. The utility distribution impacts performance. Test task accuracy and structure recovery metrics are shown from models selected on valid. task accuracy. Note that because we exclude edges to and from the closing symbol "]", recall is not equal to twice of precision for *Spanning Tree* and precision is not equal to recall for *Arborescence*.

| Model | Edge Distribution | Task Acc. | Edge Precision | Edge Recall |
|---|---|---|---|---|
| LSTM | — | $92.1 \pm 0.2$ | — | — |
| GNN on latent graph | *Indep. Undirected Edges* | $89.4 \pm 0.6$ | $20.1 \pm 2.1$ | $45.4 \pm 6.5$ |
| | *Spanning Tree* | $91.2 \pm 1.8$ | $33.1 \pm 2.9$ | $47.9 \pm 5.2$ |
| | *Indep. Directed Edges* | $90.1 \pm 0.5$ | $13.0 \pm 2.0$ | $56.4 \pm 6.7$ |
| | *Arborescence* | | | |
| GNN on latent digraph | - Neg. Exp. | $71.5 \pm 1.4$ | $23.2 \pm 10.2$ | $20.0 \pm 6.0$ |
| | - Gaussian | $\mathbf{95.0 \pm 2.2}$ | $65.3 \pm 3.7$ | $60.8 \pm 7.3$ |
| | - Gumbel | $\mathbf{95.0 \pm 3.0}$ | $\mathbf{75.5 \pm 7.0}$ | $\mathbf{71.9 \pm 12.4}$ |
| | Ground Truth Edges | $98.1 \pm 0.1$ | 100 | 100 |

*Arborescence* achieved the best performance in terms of task accuracy and structure recovery. We found that the utility distribution significantly affected performance (Table 2). For example, while negative exponential utilities induce an interpretable distribution over arborescences, App. B, we found that the multiplicative parameterization of exponentials made it difficult to train competitive models. Despite the LSTM baseline performing well on task accuracy, *Arborescence* additionally learns to recover much of the latent parse tree.

### 8.3 Learning To Explain (L2X) Aspect Ratings

With L2X we investigated the effect of the choice of relaxation. We used the BeerAdvocate dataset [57], which contains reviews comprised of free-text feedback and ratings for multiple aspects (appearance, aroma, palate, and taste; Fig. 5). Each sentence in the test set is annotated with the aspects that it describes, allowing us to define structure recovery metrics. We considered the L2X task of learning a distribution over $k$-subsets of words that best explain a given aspect rating.[8] Our model used word embeddings from [49] and convolutional neural networks with one (simple) and three (complex) layers to produce a distribution over $k$-hot binary latent masks. Given the latent masks, our model used a convolutional net to make predictions from masked embeddings. We used $k$ in $\{5, 10, 15\}$ and the following SSTs for the subset distribution: {*Euclid., Cat. Ent., Bin. Ent., E.F. Ent.*} *Top $k$* and *Corr. Top $k$*. For baselines, we used bespoke relaxations designed for this task: *L2X* [17] and *SoftSub* [86]. We trained separate models for each aspect using mean squared error (MSE).

We found that SSTs improve over bespoke relaxations (Table 3 for aspect aroma, others in App. C). For unsupervised discovery, we used the sentence-level annotations for each aspect to define ground truth subsets against which precision of the $k$-subsets was measured. SSTs tended to select subsets with higher precision across different architectures and cardinalities and achieve modest improvements in MSE. We did not find significant differences arising from the choice of regularizer $f$. Overall, the most structured SST, *Corr. Top $k$*, achieved the lowest MSE, highest precision and improved interpretability: The correlations in the model allowed it to select contiguous words, while subsets from less structured distributions were scattered (Fig. 5).

## 9 Conclusion

We introduced stochastic softmax tricks, which are random convex programs that capture a large class of relaxed distributions over structured, combinatorial spaces. We designed stochastic softmax tricks for subset selection and a variety of spanning tree distributions. We tested their use in deep latent variable models, and found that they can be used to improve performance and to encourage the unsupervised discovery of true latent structure. There are future directions in this line of work. The

Table 3 & Figure 5: For $k$-subset selection on aroma aspect, SSTs tend to outperform baseline relaxations. Test set MSE ($\times 10^{-2}$) and subset precision (%) is shown for models selected on valid. MSE. Bottom: *Corr. Top $k$* (red) selects contiguous words while *Top $k$* (blue) picks scattered words.

| Model | Relaxation | $k = 5$ | | $k = 10$ | | $k = 15$ | |
| | | MSE | Subs. Prec. | MSE | Subs. Prec. | MSE | Subs. Prec. |
|---|---|---|---|---|---|---|---|
| Simple | *L2X* [17] | $3.6 \pm 0.1$ | $28.3 \pm 1.7$ | $3.0 \pm 0.1$ | $25.5 \pm 1.2$ | $2.6 \pm 0.1$ | $25.5 \pm 0.4$ |
| | *SoftSub* [86] | $3.6 \pm 0.1$ | $27.2 \pm 0.7$ | $3.0 \pm 0.1$ | $26.1 \pm 1.1$ | $2.6 \pm 0.1$ | $25.1 \pm 1.0$ |
| | *Euclid. Top $k$* | $3.5 \pm 0.1$ | $25.8 \pm 0.8$ | $2.8 \pm 0.1$ | $32.9 \pm 1.2$ | $2.5 \pm 0.1$ | $29.0 \pm 0.3$ |
| | *Cat. Ent. Top $k$* | $3.5 \pm 0.1$ | $26.4 \pm 2.0$ | $2.9 \pm 0.1$ | $32.1 \pm 0.4$ | $2.6 \pm 0.1$ | $28.7 \pm 0.5$ |
| | *Bin. Ent. Top $k$* | $3.5 \pm 0.1$ | $29.2 \pm 2.0$ | $2.7 \pm 0.1$ | $33.6 \pm 0.6$ | $2.6 \pm 0.1$ | $28.8 \pm 0.4$ |
| | *E.F. Ent. Top $k$* | $3.5 \pm 0.1$ | $28.8 \pm 1.7$ | $2.7 \pm 0.1$ | $32.8 \pm 0.5$ | $2.5 \pm 0.1$ | $29.2 \pm 0.8$ |
| | *Corr. Top $k$* | $\mathbf{2.9 \pm 0.1}$ | $\mathbf{63.1 \pm 5.3}$ | $\mathbf{2.5 \pm 0.1}$ | $\mathbf{53.1 \pm 0.9}$ | $\mathbf{2.4 \pm 0.1}$ | $\mathbf{45.5 \pm 2.7}$ |
| Complex | *L2X* [17] | $2.7 \pm 0.1$ | $50.5 \pm 1.0$ | $2.6 \pm 0.1$ | $44.1 \pm 1.7$ | $2.4 \pm 0.1$ | $44.4 \pm 0.9$ |
| | *SoftSub* [86] | $2.7 \pm 0.1$ | $57.1 \pm 3.6$ | $\mathbf{2.3 \pm 0.1}$ | $50.2 \pm 3.3$ | $2.3 \pm 0.1$ | $43.0 \pm 1.1$ |
| | *Euclid. Top $k$* | $2.7 \pm 0.1$ | $61.3 \pm 1.2$ | $2.4 \pm 0.1$ | $52.8 \pm 1.1$ | $2.3 \pm 0.1$ | $44.1 \pm 1.2$ |
| | *Cat. Ent. Top $k$* | $2.7 \pm 0.1$ | $61.9 \pm 1.2$ | $\mathbf{2.3 \pm 0.1}$ | $52.8 \pm 1.0$ | $2.3 \pm 0.1$ | $44.5 \pm 1.0$ |
| | *Bin. Ent. Top $k$* | $2.6 \pm 0.1$ | $62.1 \pm 0.7$ | $\mathbf{2.3 \pm 0.1}$ | $50.7 \pm 0.9$ | $2.3 \pm 0.1$ | $44.8 \pm 0.8$ |
| | *E.F. Ent. Top $k$* | $2.6 \pm 0.1$ | $59.5 \pm 0.9$ | $\mathbf{2.3 \pm 0.1}$ | $54.6 \pm 0.6$ | $2.2 \pm 0.1$ | $44.9 \pm 0.9$ |
| | *Corr. Top $k$* | $\mathbf{2.5 \pm 0.1}$ | $\mathbf{67.9 \pm 0.6}$ | $\mathbf{2.3 \pm 0.1}$ | $\mathbf{60.2 \pm 1.3}$ | $\mathbf{2.1 \pm 0.1}$ | $\mathbf{57.7 \pm 3.8}$ |

Pours a slight tangerine orange and straw yellow. The head is nice and bubbly but fades very quickly with a little lacing. Smells like Wheat and European hops, a little yeast in there too. There is some fruit in there too, but you have to take a good whiff to get it. The taste is of wheat, a bit of malt, and a little fruit flavour in there too. Almost feels like drinking Champagne, medium mouthful otherwise. Easy to drink, but not something I'd be trying every night.

Appearance: 3.5    **Aroma: 4.0**    Palate: 4.5    Taste: 4.0    Overall: 4.0

relaxation framework can be generalized by modifying the constraint set or the utility distribution at positive temperatures. Some combinatorial objects might benefit from a more careful design of the utility distribution, while others, e.g., matchings, are still waiting to have their tricks designed.

## Broader Impact

This work introduces methods and theory that have the potential for improving the interpretability of latent variable models. While unfavorable consequences cannot be excluded, increased interpretability is generally considered a desirable property of machine learning models. Given that this is foundational, methodologically-driven research, we refrain from speculating further.

## Acknowledgements and Disclosure of Funding

We thank Daniel Johnson and Francisco Ruiz for their time and insightful feedback. We also thank Tamir Hazan, Yoon Kim, Andriy Mnih, and Rich Zemel for their valuable comments. MBP gratefully acknowledges support from the Max Planck ETH Center for Learning Systems. CJM is grateful for the support of the James D. Wolfensohn Fund at the Institute of Advanced Studies in Princeton, NJ. Resources used in preparing this research were provided, in part, by the Sustainable Chemical Processes through Catalysis (Suchcat) National Center of Competence in Research (NCCR), the Province of Ontario, the Government of Canada through CIFAR, and companies sponsoring the Vector Institute.

## Footnotes

[3]This is equivalent to the notion of sufficient statistics [83]. We draw a distinction only to avoid confusion, because the distributions $p_\theta$ that we ultimately consider are not necessarily from the exponential family.

[4]Convex independence is the analog of linear independence for convex combinations.

[5]For a function $f(x_1, x_2)$, $\partial f(z_1, z_2)/\partial x_1$ is the partial derivative (e.g., a gradient vector) of $f$ in the first variable evaluated at $z_1, z_2$. $df(z_1, z_2)/dx_1$ is the total derivative of $f$ in $x_1$ evaluated at $z_1, z_2$. For example, if $x = f(\theta)$, then $dg(x, \theta)/d\theta = (\partial g(x, \theta)/\partial x)(df(\theta)/d\theta) + \partial g(x, \theta)/\partial\theta$.

[6] Technically, one needs an additional local Lipschitz condition for $\mathcal{L}(X_t)$ in $\theta$ [8, Prop. 2.3, Chap. 7].

[7]We exclude edges to and from the closing symbol "]". Its edge assignments cannot be learnt from the task objective, because the correct evaluation of an operation does not depend on the closing symbol.

[8]While originally proposed for model interpretability, we used the original aspect ratings. This allowed us to use the sentence-level annotations for each aspect to facilitate comparisons between subset distributions.

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
