[Supplementary Material]

## A  Proofs for Stochastic Softmax Tricks

**Lemma 1.** *Let $\mathcal{X} \subseteq \mathbb{R}^n$ be a finite, non-empty set, $P := \mathrm{conv}(\mathcal{X})$, and $u \in \mathbb{R}^n$. We have,*

$$\max_{x \in \mathcal{X}} u^T x = \max_{x \in P} u^T x = \sup_{x \in \mathrm{relint}(P)} u^T x. \tag{9}$$

*If $\max_{x \in \mathcal{X}} u^T x$ has a unique solution $x^\star$, then $x^\star$ is also the unique solution of $\max_{x \in P} u^T x$.*

*Proof.* Assume w.l.o.g. that $\mathcal{X} = \{x_1, \ldots, x_m\}$. Let $x^\star \in \arg\max_{x \in \mathcal{X}} u^T x$.

First, let us consider the linear program over $\mathcal{X}$ vs. $P$. Clearly, $\max_{x \in \mathcal{X}} u^T x \leq \max_{x \in P} u^T x$. In the other direction, for any $y \in P$, we can write $y = \sum_i \lambda_i x_i$ for $\lambda_i \geq 0$ such that $\sum_i \lambda_i = 1$, and

$$u^T x^\star = \sum_i \lambda_i u^T x^\star \geq \sum_i \lambda_i u^T x_i = u^T y. \tag{10}$$

Hence $\max_{x \in \mathcal{X}} u^T x \geq \max_{x \in P} u^T x$. Thus $x^\star \in \arg\max_{x \in P} u^T x$.

Second, let us consider the linear program over $P$ vs. $\mathrm{relint}(P)$. The cases $x^\star \in \mathrm{relint}(P)$ or $u = 0$ are trivial, so assume otherwise. Since $u^T x^\star \geq u^T x$ for $x \in \mathrm{relint}(P)$, it suffices to show that for all $\epsilon > 0$ there exists $x_\epsilon \in \mathrm{relint}(P)$ such that $u^T x_\epsilon > u^T x^\star - \epsilon$. To that end, take $x \in \mathrm{relint}(P)$ and $0 < \lambda < \min(\epsilon, \|u\|\|x - x^\star\|)$, and define

$$x_\epsilon := x^\star + \frac{\lambda}{\|u\|\|x - x^\star\|}(x - x^\star). \tag{11}$$

$x_\epsilon \in \mathrm{relint}(P)$ by [69, Thm 6.1]. Thus, we get

$$u^T x_\epsilon = u^T x^\star + \lambda \frac{u^T(x - x^\star)}{\|u\|\|x - x^\star\|} > u^T x^\star - \epsilon \tag{12}$$

Finally, suppose that $x^\star = \arg\max_{x \in \mathcal{X}} u^T x$ is unique, but $\arg\max_{x \in P} u^T x$ contains more than just $x^\star$. We will show this implies a contradiction. Let $i^\star$ be the index $i \in \{1, \ldots, m\}$ such that $x^\star = x_{i^\star}$. Let $y \in \arg\max_{x \in P} u^T x$ be such that $y \neq x^\star$. Then we may write $y = \sum_i \lambda_i x_i$ for $\lambda_i \geq 0$ such that $\sum_i \lambda_i = 1$. But this leads to a contradiction,

$$u^T x^\star = \sum_{i \neq i^\star} \frac{\lambda_i}{1 - \lambda_{i^\star}} u^T x_i < \sum_{i \neq i^\star} \frac{\lambda_i}{1 - \lambda_{i^\star}} u^T x^\star = u^T x^\star. \tag{13}$$

$\square$

**Lemma 2.** *Let $P \subseteq \mathbb{R}^n$ be a non-empty convex polytope and $x$ an extreme point of $P$. Define the set,*

$$U(x) = \left\{ u \in \mathbb{R}^n : u^T x > u^T y, \ \forall y \in P \setminus \{x\} \right\}. \tag{14}$$

*This is the set of utility vectors of a linear program over $P$ whose argmax is the minimal face $\{x\} \subseteq P$. Then, for all $u \in U(x)$, there exists an open set $O \subseteq U(x)$ containing $u$.*

*Proof.* Let $u \in U(x)$. Let $\{x_1, \ldots, x_m\} \subseteq P$ be the set of extreme points (there are finitely many), and assume w.l.o.g. that $x = x_m$. For each $x_i \neq x_m$ there exists $\epsilon_i > 0$ such that $u^T(x_m - x_i) > \epsilon_i$. Thus, for all $v$ in the open ball $B_{r_i}(u)$ of radius $r_i = \epsilon_i / \|x_m - x_i\|$ centered at $u$, we have

$$v^T(x_m - x_i) = u^T(x_m - x_i) + (v - u)^T(x_m - x_i) > u^T(x_m - x_i) - \epsilon_i > 0. \tag{15}$$

Define $O = \cap_{i=1}^{m-1} B_{r_i}(u)$. Note, $v^T x_m > v^T x_i$ for all $v \in O, x_i \neq x_m$. Now, let $y \in P \setminus \{x_m\}$. Because $P$ is the convex hull of the $x_i$ [12, Thm. 2.9], we must have

$$y = \sum_{i=1}^m \lambda_i x_i \tag{16}$$

for $\lambda_i \geq 0, \sum_{i=1}^m \lambda_i = 1$ with at least one $\lambda_i > 0$ for $i < m$. Thus, for all $v \in O$

$$v^T x_m = \sum_{i=1}^m \lambda_i v^T x_m > \sum_{i=1}^m \lambda_i v^T x_i = v^T y. \tag{17}$$

This implies that $O \subseteq U(x_m)$, which concludes the proof, as $O$ is open, convex, and contains $u$. $\square$

**Lemma 3.** *Given a non-empty, finite set $\mathcal{X} \subseteq \mathbb{R}^n$ and a proper, closed, strongly convex function $f : \mathbb{R}^n \to \{\mathbb{R}, \infty\}$ whose domain contains the relative interior of $P := \mathrm{conv}(\mathcal{X})$, let $f^* = \min_{x \in \mathbb{R}^n} f(x)$, and $\delta_P(x)$ be the indicator function of the polytope $P$,*

$$\delta_P(x) = \begin{cases} 0 & x \in P \\ \infty & x \notin P \end{cases}. \tag{18}$$

*For $t \geq 0$, define*

$$g_t(x) := t(f(x) - f^*) + \delta_P(x), \tag{19}$$

$$g_t^*(u) := \sup_{x \in \mathbb{R}^n} u^T x - g_t(x). \tag{20}$$

*The following are true for $t > 0$,*

1. *(20) has a unique solution, $g_t^*$ is continuously differentiable, twice differentiable a.e., and*

$$\nabla g_t^*(u) = \arg\max_{x \in \mathbb{R}^n} u^T x - g_t(x). \tag{21}$$

2. *If $\max_{x \in \mathcal{X}} u^T x$ has a unique solution, then*

$$\lim_{t \to 0^+} \nabla g_t^*(u) = \arg\max_{x \in \mathcal{X}} u^T x. \tag{22}$$

*Proof.* Note, $\mathrm{relint}(P) \subseteq \mathrm{dom}(g_t) \subseteq P$.

1. Since $g_t$ is strongly convex [10, Lem. 5.20], (20) has a unique maximum [10, Thm. 5.25]. Moreover, $g_t^*$ is differentiable everywhere in $\mathbb{R}^n$ and its gradient $\nabla g_t^*$ is Lipschitz continuous [10, Thm. 5.26]. By [69, Thm 25.5] $\nabla g_t^*$ is a continuous function on $\mathbb{R}^n$. By Rademacher's theorem, $\nabla g_t^*$ is a.e. differentiable. (21) follows by standard properties of the convex conjugate [69, Thm. 23.5, Thm. 25.1].

2. First, by Lemma 1,

$$g_0^*(u) = \max_{x \in P} u^T x = \sup_{x \in \mathrm{relint}(P)} u^T x = \max_{x \in \mathcal{X}} u^T x. \tag{23}$$

Since $u$ is such that $u^T x$ is uniquely maximized over $P$, $g_0^*$ is differentiable at $u$ by [69, Thm. 23.5, Thm. 25.1]. Again by Lemma 1 we have

$$\nabla g_0^*(u) = \arg\max_{x \in P} u^T x = \arg\max_{x \in \mathcal{X}} u^T x. \tag{24}$$

Hence, our aim is to show $\lim_{t \to 0^+} \nabla g_t^*(u) = \nabla g_0^*(u)$. This is equivalent to showing that $\lim_{i \to \infty} \nabla g_{t_i}^*(u) = \nabla g_0^*(u)$ for any $t_i > 0$ such that $t_i \to 0$. Let $t_i$ be such a sequence.

We will first show that $g_{t_i}^*(u) \to g_0^*(u)$. For any $y \in \mathrm{relint}(P)$,

$$\liminf_{i \to \infty} g_{t_i}^*(u) = \lim_{i \to \infty} \inf_{j \geq i} \sup_{x \in \mathbb{R}^n} u^T x - g_{t_j}(x)$$

$$\geq \lim_{i \to \infty} \inf_{j \geq i} u^T y - g_{t_j}(y)$$

$$= u^T y$$

Thus,

$$\liminf_{i \to \infty} g_{t_i}^*(u) \geq \sup_{y \in \mathrm{relint}(P)} u^T y = g_0^*(u)$$

Since $t(f(x) - f^*) \geq 0$ for all $x \in \mathbb{R}^n$, we also have

$$\limsup_{i \to \infty} g_{t_i}^*(u) = \limsup_{i \to \infty} \sup_{x \in \mathbb{R}^n} u^T x - g_{t_i}(x)$$

$$\leq \limsup_{i \to \infty} \sup_{x \in P} u^T x = g_0^*(u).$$

Thus $\lim_{i \to \infty} g_{t_i}^*(u) = g_0^*(u)$.

By Lemma 2, there exists an open convex set $O$ containing $u$ such that for all $v \in O$, $\nabla g_0^*(u) = \arg\max_{x \in P} v^T x$. Again, $g_0^*$ is differentiable on $O$ [69, Thm. 23.5, Thm. 25.1]. Using this and the fact that $g_{t_i}^*(u) \to g_0^*(u)$, we get $\nabla g_{t_i}^*(u) \to \nabla g_0^*(u)$ [69, Thm. 25.7].

$\square$

**Proposition 1.** *If $X$ in Def. 1 is a.s. unique, then for $X_t$ in Def. 2, $\lim_{t\to 0^+} X_t = X$ a.s. If additionally $\mathcal{L}: P \to \mathbb{R}$ is bounded and continuous, then $\lim_{t\to 0^+} \mathbb{E}[\mathcal{L}(X_t)] = \mathbb{E}[\mathcal{L}(X)]$.*

*Proof.* For $g_t^*$ defined in (20), we have by Lemma 3,

$$X_t = \arg\max_{x \in P} U^T x - tf(x) = \nabla g_t^*(U). \tag{25}$$

If $X$ is a.s. unique, then again by Lemma 3

$$\begin{aligned}
\mathbb{P}\left(\lim_{t\to 0^+} X_t = X\right) &= \mathbb{P}\left(\lim_{t\to 0^+} \nabla g_t^*(U) = \arg\max_{x \in \mathcal{X}} U^T x\right) \\
&\geq \mathbb{P}\left(X \text{ is unique}\right) \\
&= 1
\end{aligned}$$

The last bit of the proof follows from the dominated convergence theorem, since the loss in bounded on $P$ by assumption, so $|\mathcal{L}(X_t)|$ is surely bounded. $\square$

**Proposition 2.** *$X_t$ in Def. 2 exists, is unique, and is a.e. differentiable and continuous in $U$.*

*Proof.* For $g_t^*$ defined in (20), we have by Lemma 3,

$$X_t = \arg\max_{x \in P} U^T x - tf(x) = \nabla g_t^*(U). \tag{26}$$

Our result follows by the other results of Lemma 3. $\square$

**Proposition 3.** *If $\mathbb{P}(U^T a = 0) = 0$ for all $a \in \mathbb{R}^n$ such that $a \neq 0$, then $X$ in Def. 1 is a.s. unique.*

*Proof.* It suffices to show that for all subsets $S \subseteq \mathcal{X}$ with $|S| > 1$, the event $\{S = \arg\max_{x \in \mathcal{X}} U^T x\}$ has zero measure. If $|S| > 1$, then we can pick two distinct points $x_1, x_2 \in S$ with $x_1 \neq x_2$. Now,

$$\mathbb{P}\left(S = \arg\max_{x \in \mathcal{X}} U^T x\right) = \mathbb{P}(\forall x \in S, U^T x = M) \leq \mathbb{P}\left(U^T(x_1 - x_2) = 0\right) = 0. \tag{27}$$

where $M = \max_{x \in \mathcal{X}} U^T x$. $\square$

**Proposition 4.** *Let $\mathcal{X} \subseteq \mathbb{R}^n$ be a non-empty finite set. If $\mathcal{X}$ is convex independent, i.e., for all $x \in \mathcal{X}$, $x \notin \text{conv}(\mathcal{X} \setminus \{x\})$, then $\mathcal{X}$ is the set of extreme points of $\text{conv}(\mathcal{X})$. In particular, any non-empty set of binary vectors $\mathcal{X} \subseteq \{0,1\}^n$ is convex independent and thus the set of extreme points of $\text{conv}(\mathcal{X})$.*

*Proof.* Let $\mathcal{X} = \{x_1, \ldots, x_m\}$. The fact that the extreme points of $\text{conv}(\mathcal{X})$ are in $\mathcal{X}$ is trivial. In the other direction, it is enough to show that $x_m$ is an extreme point. Assume $x_m \in \mathcal{X}$ is not an extreme point of $\text{conv}(\mathcal{X})$. Then by definition, we can write $x_m = \lambda y + (1-\lambda)z$ for $y, z \in \text{conv}(\mathcal{X})$, $\lambda \in (0,1)$ with $y \neq x_m$ and $z \neq x_m$. Then, we have that

$$x_m = \sum_{i=1}^{m-1} \frac{\lambda \alpha_i + (1-\lambda)\beta_i}{1 - \lambda \alpha_m - (1-\lambda)\beta_m} x_i \tag{28}$$

for some sequences $\alpha_i, \beta_i \geq 0$ such that $\sum_{i=1}^{m} \alpha_i = \sum_{i=1}^{m} \beta_i = 1$ and $\alpha_m, \beta_m < 1$. This is clearly a contradiction of our assumption that $x_m \notin \text{conv}(\mathcal{X} \setminus \{x_m\})$, since the weights in the summation (28) sum to unity. This implies that $\mathcal{X}$ are the extreme points of $\text{conv}(\mathcal{X})$.

Let $\mathcal{X} \subseteq \{0,1\}^n$. It is enough to show that $x_m \notin \text{conv}(\{x_1, \ldots, x_{m-1}\})$. Assume this is not the case. Let $c = x_m - 1/2 \in \mathbb{R}^n$, and note that $c^T x_i < c^T x_m$ for all $i \neq m$ when $x_i$ are distinct binary vectors. But, this leads to a contradiction. By assumption we can express $x_m$ as a convex combination of $x_1, \ldots, x_{m-1}$. Thus, there exists $\lambda_i \geq 0$ such that $\sum_{i=1}^{m-1} \lambda_i = 1$, and

$$c^T x_m = \sum_{i=1}^{m-1} \lambda_i c^T x_i < \sum_{i=1}^{m-1} \lambda_i c^T x_m = c^T x_m. \tag{29}$$

$\square$

# B An Abbreviated Field Guide to Stochastic Softmax Tricks

## B.1 Introduction

**Overview.** This is a short field guide to some stochastic softmax tricks (SSTs) and their associated stochastic argmax tricks (SMTs). There are many potential SSTs not discussed here. We assume throughout this Appendix that readers are completely familiar with main text and its notation; we do not review it. In particular, we follow the problem definition and notation of Section 2, the definition and notation of SMTs in Section 4, and the definition and notation of SSTs in Section 5.

This field guide is organized by the abstract set $\mathcal{Y}$. For each $\mathcal{Y}$, we identify an appropriate set $\mathcal{X} \subseteq \mathbb{R}^n$ of structured embeddings. We discuss utility distributions used in the experiments. In some cases, we can provide a simple, "closed-form", categorical sampling process for $X$, i.e., a generalization of the Gumbel-Max trick. We also cover potential relaxations used in the experiments. In the remainder of this introduction, we introduce basic concepts that recur throughout the field guide.

**Notation.** Given a finite set $S$, the indicator vector $x_T$ of a subset $T \subseteq S$ is the binary vector $x_T := (x_s)_{s \in S}$ such that $x_s = 1$ if $s \in T$ and $x_s = 0$ if $s \notin T$. For example, given an graph $G = (V, E)$, let $T$ be the edges of a spanning tree (ignoring the direction of edges). The indicator vector $x_T$ of $T$ is the vector $(x_e)_{e \in E}$ with $x_e = 1$ if $e$ is in the tree and $x_e = 0$ if $e$ is not.

$X \sim \mathcal{D}(\theta, Y)$ means that $X$ is distributed according to $\mathcal{D}$, which takes arguments $\theta$ and $Y$. Unless otherwise stated, $X$ is conditionally independent from all other random variables given $\theta, Y$. For multidimensional $U \in \mathbb{R}^n$, we use the same notation:

$$U \sim \mathrm{Exp}(\lambda) \iff U_i \sim \mathrm{Exp}(\lambda_i) \text{ independent} \tag{30}$$

Given $A \subseteq \{1, \ldots, n\}$ and $\lambda_i \in (0, \infty]$ for $0 < i \leq n$, the following notation,

$$K \sim \lambda_i \mathbf{1}_A(i), \tag{31}$$

means that $K$ is a random integer selected from $A$ with probability proportional to $\lambda_i$. If any $\lambda_i = \infty$, then we interpret this as a uniform random integer from the integers $i \in A$ with $\lambda_i = \infty$.

**Basic properties of exponentials and Gumbels.** The properties of Gumbels and exponentials are central to SMTs that have simple descriptions for the marginal $p_\theta$. We review the important ones here. These are not new; many have been used for more elaborate algorithms that manipulate Gumbels [e.g., 54].

A Gumbel random variable $G \sim \mathrm{Gumbel}(\theta)$ for $\theta \in \mathbb{R}^n$ is a location family distribution, which can be simulated using the identity

$$G \stackrel{d}{=} \theta - \log(-\log U), \tag{32}$$

for $U \sim \mathrm{uniform}(0, 1)$. An exponential random variable $E \sim \mathrm{Exp}(\lambda)$ for rate $\lambda > 0$ can be simulated using the identity

$$E \stackrel{d}{=} -\log U/\lambda, \tag{33}$$

for $U \sim \mathrm{uniform}(0, 1)$. Any result for exponentials immediately becomes a result for Gumbels, because they are monotonically related:

**Proposition 5.** *If $E \sim \mathrm{Exp}(\lambda)$, then $-\log E \sim \mathrm{Gumbel}(\log \lambda)$.*

*Proof.* If $U \sim \mathrm{uniform}(0, 1)$, then $-\log E \stackrel{d}{=} -\log(-\log U) + \log \lambda \sim \mathrm{Gumbel}(\log \lambda)$. $\qquad \square$

Although we prove results for exponentials, using their monotonic relationship, all of these results have analogs from Gumbels.

The properties of exponentials are summarized in the following proposition.

**Proposition 6.** *If $E_i \sim \mathrm{Exp}(\lambda_i)$ independent for $\lambda_i > 0$ and $i \in \{1, \ldots, n\}$, then*

    *1. $\arg\min_i E_i \sim \lambda_i$,*

    *2. $\min_i E_i \sim \mathrm{Exp}(\sum_{i=1}^n \lambda_i)$,*

3. $\min_i E_i$ *and* $\arg\min_i E_i$ *are independent,*

4. *Given* $K = \arg\min_i E_i$ *and* $E_K = \min_i E_i$, $E_i$ *for* $i \neq K$ *are conditionally, mutually independent; exponentially distributed with rates* $\lambda_i$; *and truncated to be larger than* $E_K$.

*Proof.* The joint density of $K = \arg\min_i E_i$ and $E_i$ is given by $\prod_{i=1}^n \lambda_i \exp(-\lambda_i e_i) \mathbf{1}_{x \geq e_k}(e_i)$. Manipulating this joint, we can see that

$$
\prod_{i=1}^n \lambda_i \exp(-\lambda_i e_i) \mathbf{1}_{x \geq e_k}(e_i)
$$
$$
= \lambda_k \exp(-\lambda_k e_k) \prod_{i \neq k} \lambda_i \exp(-\lambda_i e_i) \mathbf{1}_{x \geq e_k}(e_i)
$$
$$
= \frac{\lambda_k}{\sum_{i=1}^n \lambda_i} \left( \sum_{i=1}^n \lambda_i \right) \exp(-\lambda_k e_k) \prod_{i \neq k} \lambda_i \exp(-\lambda_i e_i) \mathbf{1}_{x \geq e_k}(e_i) \tag{34}
$$
$$
= \left[ \frac{\lambda_k}{\sum_{i=1}^n \lambda_i} \right] \left[ \left( \sum_{i=1}^n \lambda_i \right) \exp\left( -\sum_{i=1}^n \lambda_i e_k \right) \right] \left[ \prod_{i \neq k} \frac{\lambda_i \exp(-\lambda_i e_i)}{\exp(-\lambda_i e_k)} \mathbf{1}_{x \geq e_k}(e_i) \right]
$$

While hard to parse, this manipulation reveals the all of the assertions of the proposition. $\square$

Prop. 6 has a couple of corollaries. First, subtracting the minimum exponential from a collection only affects the distribution of the minimum, leaving the distribution of the other exponentials unchanged.

**Corollary 1.** *If* $E_i \sim \mathrm{Exp}(\lambda_i)$ *independent for* $\lambda_i > 0$ *and* $i \in \{1, \dots, n\}$, *then* $E_i - \min_i E_i$ *are mutually independent and*

$$
E_i - \min_i E_i \sim \begin{cases} \mathrm{Exp}(\lambda_i) & i \neq K \\ 0 & i = K \end{cases}, \tag{35}
$$

*where* $K = \arg\min_i E_i$.

*Proof.* Consider the change of variables $e_i' = e_i - e_k$ in the joint (34). Each of the terms in the right hand product over $i \neq k$ of (34) are transformed in the following way

$$
\frac{\lambda \exp(-\lambda_i (e_i' + e_k))}{\exp(-\lambda_i e_k)} \mathbf{1}_{x \geq e_k}(e_i' + e_k) \longrightarrow \lambda_i \exp(-\lambda_i e_i') \tag{36}
$$

This is essentially the memoryless property of exponentials. Thus, the $E_i' = E_i - E_K$ for $i \neq K$ are distributed as exponentials with rate $\lambda_i$ and mutually independent. $E_K'$ is the constant 0, which is independent of any random variable. Our result follows. $\square$

Second, the process of sorting the collection $E_i$ is equivalent to sampling from $\{1, \dots, n\}$ without replacement with probabilities proportional to $\lambda_i$.

**Corollary 2.** *Let* $E_i \sim \mathrm{Exp}(\lambda_i)$ *independent for* $\lambda_i > 0$ *and* $i \in \{1, \dots, n\}$. *Let* $\mathrm{argsort}_x : \{1, \dots, n\} \to \{1, \dots, n\}$ *be the argsort permutation of* $x \in \mathbb{R}^n$, *i.e., the permutation such that* $x_{\mathrm{argsort}_x(i)}$ *is in non-decreasing order. We have*

$$
\mathbb{P}(\mathrm{argsort}_E = \sigma) = \prod_{i=1}^n \frac{\lambda_{\sigma(i)}}{\sum_{j=i}^n \lambda_{\sigma(j)}} \tag{37}
$$

*Given* $\mathrm{argsort}_E = \sigma$, *the sorted vector* $E_\sigma = (E_{\sigma(i)})_{i=1}^n$ *has the following distribution,*

$$
E_{\sigma(1)} \sim \mathrm{Exp}\left( \sum_{j=1}^n \lambda_{\sigma(j)} \right)
$$
$$
E_{\sigma(i)} - E_{\sigma(i-1)} \sim \mathrm{Exp}\left( \sum_{j=i}^n \lambda_{\sigma(j)} \right) \tag{38}
$$

*Proof.* This follows after repeated, interleaved uses of Cor. 1 and Prop. 6. $\square$

## B.2 Element Selection

**One-hot binary embeddings.** Given a finite set $\mathcal{Y}$ with $|\mathcal{Y}| = n$, we can associate each $y \in \mathcal{Y}$ with a one-hot binary embedding. Let $\mathcal{X} \subseteq \mathbb{R}^n$ be the following set of one-hot embeddings,

$$\mathcal{X} = \left\{ x \in \{0, 1\}^n \,\middle|\, \sum_i x_i = 1 \right\}. \tag{39}$$

For $u \in \mathbb{R}^n$, a solution to the linear program $x^\star \in \arg\max_{x \in \mathcal{X}} u^T x$ is given by setting $x_k^\star = 1$ for $k \in \arg\max_i u_i$ and $x_k^\star = 0$ otherwise.

**Random Utilities.** If $U \sim \mathrm{Gumbel}(\theta)$, then $X \sim \exp(\theta_i)$. This is known as the Gumbel-Max trick [52, 54], which follows from Props. 5 and 6.

**Relaxtions.** If $f(x) = \sum_i x_i \log x_i$, then the SST solution $X_t$ is given by

$$X_t = \left( \frac{\exp(U_i/t)}{\sum_{j=1}^n \exp(U_j/t)} \right)_{i=1}^n. \tag{40}$$

In this case, the categorical entropy relaxation and the exponential family relaxation coincide. This is known as the Gumbel-Softmax trick when $U \sim \mathrm{Gumbel}(\theta)$ [53, 35]. If $f(x) = \|x\|^2/2$, then $X_t$ can be computed using the `sparsemax` operator [55]. In analogy, we name this relaxation with $U \sim \mathrm{Gumbel}(\theta)$ the Gumbel-Sparsemax trick.

## B.3 Subset Selection

**Binary vector embeddings.** Given a finite set $S$ with $|S| = n$, let $\mathcal{Y}$ be the set of all subsets of $S$, i.e., $\mathcal{Y} = 2^S := \{y \subseteq S\}$. The indicator vector embeddings of $\mathcal{Y}$ is the set,

$$\mathcal{X} = \{x_y : y \in 2^S\} = \{0, 1\}^{|S|} \tag{41}$$

For $u \in \mathbb{R}^n$, a solution to the linear program $x^\star \in \arg\max_{x \in \mathcal{X}} u^T x$ is given by setting $x_i^\star = 1$ if $u_i > 0$ and $x_i^\star = 0$ otherwise, for all $i \leq n$.

**Random utilities.** If $U \sim \mathrm{Logistic}(\theta)$, then $X_i \sim \mathrm{Bern}(\sigma(\theta_i))$ for all $i \leq n$, where $\sigma(\cdot)$ is the sigmoid function. This corresponds to an application of the Gumbel-Max trick independently to each element in $S$. $U \sim \mathrm{Logistic}(\theta)$ has the same distribution as $\theta + \log U' - \log(1 - U')$ for $U' \sim \mathrm{uniform}(0, 1)$.

**Relaxations.** For this case, the exponential family and the binary entropy relaxation, where $f(x) = \sum_{i=1}^n x_i \log(x_i) + (1 - x_i) \log(1 - x_i)$, coincide. The SST solution $X_t$ is given by

$$X_t = (\sigma(U_i/t))_{i=1}^n \tag{42}$$

where $\sigma(\cdot)$ is the sigmoid function. For the categorical entropy relaxation with $f(x) = \sum_{i=1}^n x_i \log(x_i)$, the SST solution is given by $X_t = (\min(1, \exp(U_i/t)))_{i=1}^n$ [13].

## B.4 $k$-Subset Selection

$k$**-hot binary embeddings.** Given a finite set $S$ with $|S| = n$, let $\mathcal{Y}$ be the set of all subsets of $S$ with cardinality $1 \leq k < n$, i.e., $\mathcal{Y} = \{y \subseteq S \,|\, |y| = k\}$. The indicator vector embeddings of $\mathcal{Y}$ is the set,

$$\mathcal{X} = \{x_y : y \subseteq S, \, |y| = k\} \tag{43}$$

For $u \in \mathbb{R}^n$, let $\arg\mathrm{topk}\, u$ be the operator that returns the indices of the $k$ largest values of $u$. For $u \in \mathbb{R}^n$, a solution to the linear program $x^\star \in \arg\max_{x \in \mathcal{X}} u^T x$ is given by setting $x_i^\star = 1$ for $i \in \arg\mathrm{topk}\, u$ and $x_i^\star = 0$ otherwise.

**Random utilities.** If $U \sim \mathrm{Gumbel}(\theta)$, this induces a Plackett-Luce model [52][66] over the indices that sort $U$ in descending order. In particular, $X$ may be sampled by sampling $k$ times without replacement from the set $\{1, \ldots, n\}$ with probabilities proportional to $\exp(\theta_i)$, setting the sampled indices of $X$ to 1, and the rest to 0 [44]. This can be seen as a consequence of Cor. 2.

**Relaxations.** For the Euclidean relaxation with $f(x) = \|x\|^2/2$, $X_t$ we computed $X_t$ using a bisection method to solve the constrained quadratic program, but note that other algorithms are available [13]. For the categorical entropy relaxation with $f(x) = \sum_{i=1}^{n} x_i \log(x_i)$, the SST solution $X_t$ can be computed efficiently using the algorithm described in [56]. For the binary entropy relaxation with $f(x) = \sum_{i=1}^{n} x_i \log(x_i) + (1-x_i)\log(1-x_i)$, the SST solution can be computed using the algorithm in [7]. Finally, for the exponential family relaxation, the SST solution can be computed using dynamic programming as described in [79].

## B.5 Correlated $k$-Subset Selection

**Correlated $k$-hot binary embeddings.** Given a finite set $S$ with $|S| = n$, let $\mathcal{Y}$ be the set of all subsets of $S$ with cardinality $1 \leq k < n$, i.e., $\mathcal{Y} = \{y \subseteq V \,|\, |y| = k\}$. We can associate each $y \in \mathcal{Y}$ with a $(2n-1)$-dimensional binary embedding with a $k$-hot cardinality constraint on the first $n$ dimensions and a constraint that the $n-1$ dimensions indicate correlations between adjacent dimensions in the first $n$, i.e. the vertices of the correlation polytope of a chain [83, Ex. 3.8] with an added cardinality constraint [59]. Let $\mathcal{X} \subseteq \mathbb{R}^n$ be the set of all such embeddings,

$$\mathcal{X} = \left\{ x \in \{0,1\}^{2n-1} \,\middle|\, \sum_{i=1}^{n} x_i = k; \; x_i = x_{i-n} x_{i-n+1} \text{ for all } n < i \leq 2n-1 \right\}. \qquad (44)$$

For $u \in \mathbb{R}^n$, a solution to the linear program $x^\star \in \arg\max_{x \in \mathcal{X}} u^T x$ can be computed using dynamic programming [79, 59].

**Random utilities.** In our experiments for correlated $k$-subset selection we considered Gumbel unary utilities with fixed pairwise utilities. This is, we considered $U_i \sim \mathrm{Gumbel}(\theta_i)$ for $i \leq n$ and $U_i = \theta_i$ for $n < i \leq 2n-1$.

**Relaxations.** The exponential family relaxation for correlated $k$-subsets can be computed using dynamic programming as described in [79, 59].

## B.6 Perfect Bipartite Matchings

**Permutation matrix embeddings.** Given a complete bipartite graph $K_{n,n}$, let $\mathcal{Y}$ be the set of all perfect matchings. We can associate each $y \in \mathcal{Y}$ with a permutation matrix and let $\mathcal{X}$ be the set of all such matrices,

$$\mathcal{X} = \left\{ x \in \{0,1\}^{n \times n} \,\middle|\, \text{ for all } 1 \leq i,j \leq n, \; \sum_i x_{ij} = 1, \; \sum_j x_{ij} = 1 \right\}. \qquad (45)$$

For $u \in \mathbb{R}^{n \times n}$, a solution to the linear program $x^\star \in \arg\max_{x \in \mathcal{X}} u^T x$ can be computed using the Hungarian method [47].

**Random utilities.** Previously, [58] considered $U \sim \mathrm{Gumbel}(\theta)$ and [31] uses correlated Gumbel-based utilities that induce a Plackett-Luce model [52][66].

**Relaxations.** For the categorical entropy relaxation with $f(x) = \sum_{i=1}^{n} x_i \log(x_i)$, the SST solution $X_t$ can be computed using the Sinkhorn algorithm [75]. When choosing Gumbel utilities, this recovers Gumbel-Sinkhorn [58]. This relaxation can also be used to relax the Plackett-Luce model, if combined with the utility distribution in [31].

## B.7 Undirected Spanning Trees

**Edge indicator embeddings.** Given a undirected graph $G = (V, E)$, let $\mathcal{Y}$ be the set of spanning trees of $G$ represented as subsets $T \subseteq E$ of edges. The indicator vector embeddings of $\mathcal{Y}$ is the set,

$$\mathcal{X} = \cup_{T \in \mathcal{Y}} \{x_T\}. \qquad (46)$$

We assume that $G$ has at least one spanning tree, and thus $\mathcal{X}$ is non-empty. A linear program over $\mathcal{X}$ is known as a maximum weight spanning tree problem. It is efficiently solved by the Kruskal's algorithm [46].

**Random utilities.**   In our experiments, we used $U \sim \text{Gumbel}(\theta)$. In this case, there is a simple, categorical sampling process that described the distribution over $X$.

The sampling process follows Kruskal's algorithm [46]. The steps of Kruskal's algorithm are as follows: sort the list of edges $e$ in non-increasing order according to their utilities $U_e$, greedily construct a tree by adding edges to $T$ as long as no cycles are created, and return the indicator vector $x_T$. Using Cor. 2 and Prop. 5, for Gumbel utilities this is equivalent to the following process: sample edges $e$ without replacement with probabilities proportional to $\exp(\theta_e)$, add edges $e$ to $T$ in the sampled order as long as no cycles are created, and return the indicator vector $x_T$.

**Relaxations.**   The exponential family relaxation for spanning trees can be computed using Kirchhoff's Matrix-Tree Theorem. Here we present a quick informal review. Consider an exponential family with natural parameters $u \in \mathbb{R}^{|E|}$ over $\mathcal{X}$ such that the probability of $x \in \mathcal{X}$ is proportional to $\exp(u^T x)$. Define the weights,

$$w_{ij} = \begin{cases} \exp(u_e) & \text{if } i \neq j \text{ and } \exists\, e \in E \text{ connecting nodes } i \text{ and } j \\ 0 & \text{otherwise} \end{cases}. \tag{47}$$

Consider the graph Laplacian $L \in \mathbb{R}^{|V| \times |V|}$ defined by

$$L_{ij} = \begin{cases} \sum_{k \neq j} w_{kj} & \text{if } i = j \\ -w_{ij} & \text{if } i \neq j \end{cases} \tag{48}$$

Let $L^{k,k}$ be the submatrix of $L$ obtained by deleting the $k$th row and $k$th column. The Kirchhoff Matrix-Tree Theoroem states that

$$\log \det L^{k,k} = \log \left( \sum_{T \in \mathcal{Y}} \exp\left(u^T x_T\right) \right). \tag{49}$$

[82, p. 14] for a reference. We can use this to compute the marginals of the exponential family via its derivative [83]. In particular,

$$\mu(u) := \left( \frac{\partial \log \det L^{k,k}}{\partial u_e} \right)_{e \in E} = \sum_{T \in \mathcal{Y}} \frac{x_T \exp\left(u^T x_T\right)}{\sum_{T' \in \mathcal{Y}} \exp\left(u^T x_{T'}\right)}. \tag{50}$$

These partial derivatives can be computed in the standard auto-diff libraries. All together, we may define the exponential family relaxation via $X_t = \mu(U/t)$.

### B.8   Rooted, Directed Spanning Trees

**Edge indicator embeddings.**   Given a directed graph $G = (V, E)$, let $\mathcal{Y}$ be the set of $r$-arborescences for $r \in V$. An $r$-arborescence is a subgraph of $G$ that is a spanning tree if the edge directions are ignored and that has a directed path from $r$ to every node in $V$. Let $x_T := (x_e)_{e \in E}$ be the indicator vector of an $r$-arborescence with edges $T \subseteq E$. Define the set $\mathcal{T}(r)$ of $r$-arborescences of $G$. The indicator vector embeddings of $\mathcal{Y}$ is the set,

$$\mathcal{X} = \cup_{T \in \mathcal{T}(r)} \{x_T\}. \tag{51}$$

We assume that $G$ has at least one $r$-arborescence, and thus $\mathcal{X}$ is non-empty. A linear program over $\mathcal{X}$ is known as a maximum weight $r$-arborescence problem. It is efficiently solved by the Chu-Liu-Edmonds algorithm (CLE) [18, 24], see Alg. 1 for an implementation by [39].

**Random utilities.**   In the experiments, we tried $U \sim \text{Gumbel}(\theta)$, $-U \sim \text{Exp}(\theta)$ with $\theta > 0$, and $U \sim \mathcal{N}(\theta, 1)$. As far as we know $X$ does not have any particularly simple closed-form categorical sampling process in the cases $U \sim \text{Gumbel}(\theta)$ or $U \sim \mathcal{N}(\theta, 1)$.

In contrast, for negative exponential utilities $-U \sim \text{Exp}(\theta)$, $X$ can be sampled using the sampling process given in Alg. 2. In some sense, Alg. 2 is an elaborate generalization of the Gumbel-Max trick to arborescences.

We will argue that Alg. 2 produces the same distribution over its output as Alg. 1 does on negative exponential $U_e$. To do this, we will argue that joint distribution of the sequence of edge choices (lines

| **Algorithm 1:** Maximum $r$-arborescence [39] | **Algorithm 2:** Equiv. for neg. exp. $U$ |
|---|---|
| **Init:** graph $G$, node $r$, $U_e \in \mathbb{R}$, $T = \emptyset$; | **Init:** graph $G$, node $r$, $\lambda_e > 0$, $T = \emptyset$; |
| 1 **foreach** *node* $v \neq r$ **do** | 1 **foreach** *node* $v \neq r$ **do** |
| 2     $E_v = \{\text{edges entering } v\}$; | 2     $E_v = \{\text{edges entering } v\}$; |
| 3     $U'_e = U_e - \max_{e \in E_v} U_e, \forall e \in E_v$; | 3     Sample $e \sim \lambda_e \mathbf{1}_{E_v}(e)$; $T = T \cup \{e\}$; |
| 4     Pick $e \in E_v$ s.t. $U'_e = 0$; $T = T \cup \{e\}$; | 4     $\lambda'_a = \lambda_a$ if $a \neq e$ else $\infty$, $\forall a \in E_v$; |
| 5 **if** $T$ *is an arborescence* **then return** $x_T$; | 5 **if** $T$ *is an arborescence* **then return** $x_T$; |
| 6 **else** there is a directed cycle $C \subseteq T$ | 6 **else** there is a directed cycle $C \subseteq T$ |
| 7     Contract $C$ to supernode, form graph $G'$; | 7     Contract $C$ to supernode, form graph $G'$; |
| 8     Recurse on $(G', r, U')$ to get arbor. $T'$; | 8     Recurse on $(G', r, \lambda')$ to get arbor. $T'$; |
| 9     Expand $T'$ to subgraph of $G$ and add | 9     Expand $T'$ to subgraph of $G$ and add |
| 10     all but one edge of $C$; **return** $x_{T'}$; | 10     all but one edge of $C$; **return** $x_{T'}$; |

Figure 6: Alg. 1 and Alg. 2 have the same output distribution for negative exponential $U$, i.e., Alg. 2 is an equivalent categorical sampling process for $X$. Alg. 1 computes the maximum point of a stochastic $r$-arborescence trick with random utilities $U_e$ [39]. When $-U_e \sim \text{Exp}(\lambda_e)$, it has the same distribution as Alg. 2. Alg. 2 samples a random $r$-arborescence given rates $\lambda_e > 0$ for each edge. Both Algs. assume that $G$ has at least one $r$-arbor. Color indicates the main difference.

2-4 colored red in Alg. 1), after integrating out $U$, is given by lines 2-4 (colored blue) of Alg. 2. Consider the first call to CLE: all $U_e$ are negative and distinct almost surely, for each node $v \neq r$ the maximum utility edge is picked from the set of entering edges $E_v$, and all edges have their utilities modified by subtracting the maximum utility. The argmax of $U_e$ over $E_v$ is a categorical random variable with mass function proportional to the rates $\lambda_e$, and it is independent of the max of $U_e$ over $E_v$ by Prop. 6. By Cor. 1, the procedure of modifying the utilities leaves the distribution of all unpicked edges invariant and sets the utility of the argmax edge to 0. Thus, the distribution of $U'$ passed one level up the recursive stack is the same as $U$ with the exception of a randomly chosen subset of utilities $U'_e$ whose rates have been set to $\infty$. The equivalence in distribution between Alg. 1 and Alg. 2 follows by induction.

**Relaxations.** The exponential family relaxation for $r$-arborescences can be computed using the directed version of Kirchhoff's Matrix-Tree Theorem. Here we present a quick informal review. Consider an exponential family with natural parameters $u \in \mathbb{R}^{|E|}$ over $\mathcal{X}$ such that the probability of $x \in \mathcal{X}$ is proportional to $\exp(u^T x)$. Define the weights,

$$w_{ij} = \begin{cases} \exp(u_e) & \text{if } i \neq j \text{ and } \exists\, e \in E \text{ from node } i \to j \\ 0 & \text{otherwise} \end{cases}. \tag{52}$$

Consider the graph Laplacian $L \in \mathbb{R}^{|V| \times |V|}$ defined by

$$L_{ij} = \begin{cases} \sum_{k \neq j} w_{kj} & \text{if } i = j \\ -w_{ij} & \text{if } i \neq j \end{cases} \tag{53}$$

Let $L^{r,r}$ be the submatrix of $L$ obtained by deleting the $r$th row and $r$th column. The result by Tutte [82, p. 140] states that

$$\log \det L^{r,r} = \log \left( \sum_{T \in \mathcal{T}(r)} \exp\left( u^T x_T \right) \right) \tag{54}$$

We can use this to compute the marginals of the exponential family via its derivative [83]. In particular,

$$\mu(u) := \left( \frac{\partial \log \det L^{r,r}}{\partial u_e} \right)_{e \in E} = \sum_{T \in \mathcal{T}(r)} \frac{x_T \exp\left( u^T x_T \right)}{\sum_{T' \in \mathcal{T}(r)} \exp\left( u^T x_{T'} \right)}. \tag{55}$$

These partial derivatives can be computed in the standard auto-diff libraries. All together, we may define the exponential family relaxation via $X_t = \mu(U/t)$.

Table 4: For $k$-subset selection on appearance aspect, SSTs select subsets with high precision and outperform baseline relaxations. Test set MSE ($\times 10^{-2}$) and subset precision (%) is shown for models selected on valid. MSE.

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

## C   Additional Results

### C.1   REINFORCE and NVIL on Graph Layout

We experimented with 3 variants of REINFORCE estimators, each with a different baseline. The *EMA* baseline is an exponential moving average of the ELBO. The *Batch* baseline is the mean ELBO of the current batch. Finally, the *Multi-sample* baseline is the mean ELBO over $k$ multiple samples, which is a local baseline for each sample (See section 3.1 of [43]). For NVIL, the input-dependent baseline was a one hidden-layer MLP with ReLU activations, attached to the GNN encoder, just before the final fully connected layer. We did not do variance normalization. We used weight decay on the encoder parameters, including the input-dependent baseline parameters. We tuned weight decay and the exponential moving average constant, in addition to the learning rate. For *Multi-sample* REINFORCE, we additionally tuned $k = \{2, 4, 8\}$, and following [43], we divided the batch size by $k$ in order to keep the number of total samples constant.

Table 6: For $k$-subset selection on taste aspect, MSE and subset precision tend to be lower for all methods. This is because the taste rating is highly correlated with other ratings making it difficult to identify subsets with high precision. SSTs achieve small improvements. Test set MSE ($\times 10^{-2}$) and subset precision (%) is shown for models selected on valid. MSE.

| Model | Relaxation | $k=5$ MSE | Subs. Prec. | $k=10$ MSE | Subs. Prec. | $k=15$ MSE | Subs. Prec. |
|---|---|---|---|---|---|---|---|
| Simple | *L2X* [17] | $3.1 \pm 0.1$ | $28.5 \pm 0.6$ | $2.9 \pm 0.1$ | $24.1 \pm 1.3$ | $2.7 \pm 0.1$ | $26.8 \pm 0.8$ |
| | *SoftSub* [86] | $3.1 \pm 0.1$ | $29.9 \pm 0.8$ | $2.9 \pm 0.1$ | $27.7 \pm 0.7$ | $2.7 \pm 0.1$ | $27.8 \pm 1.9$ |
| | *Euclid. Top k* | $3.0 \pm 0.1$ | $30.2 \pm 0.4$ | $2.7 \pm 0.1$ | $28.0 \pm 0.4$ | $2.6 \pm 0.1$ | $26.5 \pm 0.5$ |
| | *Cat. Ent. Top k* | $3.1 \pm 0.1$ | $28.5 \pm 0.6$ | $2.8 \pm 0.1$ | $28.9 \pm 0.6$ | $2.6 \pm 0.1$ | $30.5 \pm 1.6$ |
| | *Bin. Ent. Top k* | $3.0 \pm 0.1$ | $29.2 \pm 0.4$ | $2.9 \pm 0.1$ | $24.6 \pm 1.7$ | $2.6 \pm 0.1$ | $27.9 \pm 0.9$ |
| | *E.F. Ent. Top k* | $3.0 \pm 0.1$ | $29.7 \pm 0.3$ | $2.7 \pm 0.1$ | $29.0 \pm 1.5$ | $2.6 \pm 0.1$ | $26.5 \pm 0.5$ |
| | *Corr. Top k* | $\mathbf{2.8 \pm 0.1}$ | $\mathbf{31.7 \pm 0.5}$ | $\mathbf{2.5 \pm 0.1}$ | $\mathbf{37.7 \pm 1.6}$ | $\mathbf{2.4 \pm 0.1}$ | $\mathbf{37.8 \pm 0.5}$ |
| Complex | *L2X* [17] | $2.5 \pm 0.1$ | $40.3 \pm 0.7$ | $2.4 \pm 0.1$ | $42.4 \pm 2.0$ | $2.4 \pm 0.1$ | $39.7 \pm 1.1$ |
| | *SoftSub* [86] | $2.5 \pm 0.1$ | $43.3 \pm 0.9$ | $2.4 \pm 0.1$ | $41.3 \pm 0.5$ | $2.3 \pm 0.1$ | $40.5 \pm 0.7$ |
| | *Euclid. Top k* | $\mathbf{2.4 \pm 0.1}$ | $43.8 \pm 0.7$ | $2.3 \pm 0.1$ | $43.1 \pm 0.6$ | $2.2 \pm 0.1$ | $42.2 \pm 0.8$ |
| | *Cat. Ent. Top k* | $\mathbf{2.4 \pm 0.1}$ | $\mathbf{46.5 \pm 0.6}$ | $2.3 \pm 0.1$ | $44.6 \pm 0.3$ | $2.2 \pm 0.1$ | $45.5 \pm 1.1$ |
| | *Bin. Ent. Top k* | $\mathbf{2.4 \pm 0.1}$ | $40.9 \pm 1.3$ | $2.3 \pm 0.1$ | $46.3 \pm 0.9$ | $2.2 \pm 0.1$ | $44.7 \pm 0.5$ |
| | *E.F. Ent. Top k* | $\mathbf{2.4 \pm 0.1}$ | $45.3 \pm 0.6$ | $\mathbf{2.2 \pm 0.1}$ | $46.1 \pm 0.8$ | $2.2 \pm 0.1$ | $\mathbf{46.6 \pm 1.1}$ |
| | *Corr. Top k* | $\mathbf{2.4 \pm 0.1}$ | $45.9 \pm 1.3$ | $\mathbf{2.2 \pm 0.1}$ | $\mathbf{47.3 \pm 0.6}$ | $\mathbf{2.1 \pm 0.1}$ | $45.1 \pm 2.0$ |

Table 7: NVIL and REINFORCE struggle to learn the underlying structure wherever their SST counterparts struggle. NVIL with *Spanning Tree* is able to learn some structure, but it is still worse and higher variance than its SST counterpart. This is for $T = 10$.

| Edge Distribution | REINFORCE (*EMA*) ELBO | Edge Prec. | Edge Rec. | NVIL ELBO | Edge Prec. | Edge Rec. |
|---|---|---|---|---|---|---|
| *Indep. Directed Edges* | $-1730 \pm 60$ | $41 \pm 4$ | $92 \pm 7$ | $-1550 \pm 20$ | $44 \pm 1$ | $94 \pm 1$ |
| *Top $|V| - 1$* | $-2170 \pm 10$ | $42 \pm 1$ | $42 \pm 1$ | $-2110 \pm 10$ | $42 \pm 2$ | $42 \pm 2$ |
| *Spanning Tree* | $-2250 \pm 20$ | $40 \pm 7$ | $40 \pm 7$ | $-1570 \pm 300$ I | $83 \pm 20$ | $83 \pm 20$ |

| Edge Distribution | REINFORCE (*Batch*) ELBO | Edge Prec. | Edge Rec. | REINFORCE (*Multi-sample*) ELBO | Edge Prec. | Edge Rec. |
|---|---|---|---|---|---|---|
| *Indep. Directed Edges* | $-1780 \pm 20$ | $39 \pm 3$ | $90 \pm 6$ | $-1710 \pm 30$ | $38 \pm 3$ | $88 \pm 6$ |
| *Top $|V| - 1$* | $-2180 \pm 0$ | $39 \pm 1$ | $39 \pm 1$ | $-2150 \pm 10$ | $40 \pm 0$ | $40 \pm 0$ |
| *Spanning Tree* | $-2260 \pm 0$ | $41 \pm 1$ | $41 \pm 1$ | $-2230 \pm 20$ | $42 \pm 1$ | $42 \pm 1$ |

We used $U$ as the "action" for all edge distributions, and therefore, computed the log probability over $U$. We also computed the KL divergence with respect to $U$ as in the rest of the graph layout experiments (See App. D.5.2).This was because computing the probability of $X$ is not computationally efficient for *Top $|V| - 1$* and *Spanning Tree*. In particular, the marginal of $X$ in these cases is not in the exponential family. We emphasize that using $U$ as the "action" for REINFORCE is atypical.

We found that both NVIL and REINFORCE with *Indep. Directed Edges* and *Top $|V| - 1$* perform similarly to their SST counterparts, struggling to learn the underlying structure. This is also the case for REINFORCE with *Spanning Tree*. On the other hand, NVIL with *Spanning Tree*, is able to learn some structure, although worse and higher variance than its SST counterpart.

# D  Experimental Details

## D.1  Implementing Relaxed Gradient Estimators

For implementing the relaxed gradient estimator given in (7), several options are available. In general, the forward computation of $X_t$ may be unrolled, such that the estimator can be computed with the aid of modern software packages for automatic differentiation [1, 65, 16]. However, for some specific choices of $f$ and $\mathcal{X}$, it may be more efficient to compute the estimator exactly via a custom backward

pass, e.g. [7, 56]. Yet another alternative is to use local finite difference approximations as pointed out by [21]. In this case, an approximation for $d\mathcal{L}(X_t)/dU$ is given by

$$\frac{d\mathcal{L}(X_t)}{dU} \approx \frac{X_t(U + \epsilon\partial\mathcal{L}(X_t)/\partial X_t) - X_t(U - \epsilon\partial\mathcal{L}(X_t)/\partial X_t)}{2\epsilon} \qquad (56)$$

with equality in the limit as $\epsilon \to 0$. This approximation is valid, because the Jacobian of $X_t$ is symmetric [70, Cor. 2.9]. It is derived from the vector chain rule and the definition of the derivative of $X_t$ in the direction $\partial\mathcal{L}(X_t)/\partial X_t$. This method only requires two additional calls to a solver for (6) and does not require additional evaluations of $\mathcal{L}$. We found this method helpful for implementing *E.F. Ent. Top $k$* and *Corr. Top $k$*.

### D.2   Numerical Stability

Our SSTs for undirected and rooted direct spanning trees (*Spanning Tree* and *Arborescence*) require the inversion of a matrix. We found matrix inversion prone to suffer from numerical instabilities when the maximum and minimum values in $\theta$ grew too large apart. As a resolution, we found it effective to cap the maximal range in $\theta$ to 15 during training. Specifically, if $\theta_{\max} = \max(\theta)$, we clipped, i.e., $\theta_i = \max(\theta_i, \theta_{\max} - 15)$. In addition, after clipping we normalized, i.e., $\theta = \theta - \theta_{\max}$. This leaves the computation unaffected but improves stability. In addition, for *Spanning Tree* we chose the index $k$ (c.f., Section B) to be the row in which $\theta_{\max}$ occurs. We did not clip when evaluating the models.

### D.3   Estimating Standard Errors by Bootstrapping Model Selection

For all our experiments, we report standard errors over the model selection process from bootstrapping. In all our experiments we randomly searched hyperparameters over $N = 20$ (NRI, ListOps) or $N = 25$ (L2X) independent runs and selected the best model over these runs based on the task objective on the validation set. In all tables, we report test set metrics for the best model thus selected. We obtained standard errors by bootstrapping this procedure. Specifically, we randomly sampled with replacement $N$ times from the $N$ runs and selected models on the sampled runs. We repeated this procedure for $M = 10^5$ times to compute standard deviations for all test set metrics over the $M$ trials.

### D.4   Computing the KL divergence

There are at least 3 possible KL terms for a relaxed ELBO: the KL from a prior over $\mathcal{X}$ to the distribution of $X$, the KL from a prior over $\mathrm{conv}(\mathcal{X})$ to the distribution of $X_t$, or the KL from a prior over $\mathbb{R}^n$ to the distribution of $U$, see Section C.3 of [53] for a discussion of this topic. In our case, since we do not know of an explicit tractable density of $X_t$ or $X$, we compute the KL with respect to $U$. The KL divergence with respect to $U$ is an *upper-bound* to the KL divergence with respect to $X_t$ due to a data processing inequality. Therefore, the ELBO that we are optimizing is a *lower-bound* to the relaxed variational objective. Whether or not this is a good choice is an empirical question. Note, that *when optimizing the relaxed objective*, using a KL divergence with respect to $X$ does not result in a lower-bound to the relaxed variational objective, as it is not necessarily an ELBO for the continuous relaxed model (see again Section C.3 of [53]).

### D.5   Neural Relational Inference (NRI) for Graph Layout

#### D.5.1   Data

Our dataset consisted of latent prior spanning trees over 10 vertices. Latent spanning trees were sampled by applying Kruskal's algorithm [46] to $U \sim \mathrm{Gumbel}(0)$ for a fully-connected graph. Note that this does not result in a uniform distribution over spanning trees. Initial vertex locations were sampled from $\mathcal{N}(0, I)$ in $\mathbb{R}^2$. Given initial locations and the latent tree, dynamical observations were obtained by applying a force-directed algorithm for graph layout [25] for $T \in \{10, 20\}$ iterations. We then discarded the initial vertex positions, because the first iteration of the layout algorithm typically results in large relocations. This renders the initial vertex positions an outlier which is hard to model. Hence, the final dataset used for training consisted of 10 respectively 20 location observations in $\mathbb{R}^2$ for each of the 10 vertices. By this procedure, we generated a training set of size 50,000 and validation and test sets of size 10,000.

### D.5.2 Model

The NRI model consists of encoder and decoder graph neural networks. Our encoder and decoder architectures were identical to the MLP encoder and MLP decoder architectures, respectively, in [38].

**Encoder**   The encoder GNN passes messages over the fully connected directed graph with $n = 10$ nodes. We took the final edge representation of the GNN to use as $\theta$. The final edge representation was in $\mathbb{R}^{90 \times m}$, where $m = 2$ for *Indep. Directed Edges* and $m = 1$ for *E.F. Ent. Top* $|V| - 1$ and *Spanning Tree*, both over undirected edges (90 because we considered all directed edges excluding self-connections). We had $m = 2$ for *Indep. Directed Edges*, because we followed [38] and applied the Gumbel-Max trick independently to each edge. This is equivalent to using $U \sim \text{Logistic}(\theta)$, where $\theta \in \mathbb{R}^{90}$. Both *E.F. Ent. Top* $|V| - 1$ and *Spanning Tree* require undirected graphs, therefore, we "symmetrized" $\theta$ such that $\theta_{ij} = \theta_{ji}$ by taking the average of the edge representations for both directions. Therefore, in this case, $\theta \in \mathbb{R}^{45}$.

**Decoder**   Given previous timestep data, the decoder GNN passes messages over the sampled graph adjacency matrix $X$ and predicts future node positions. As in [38], we used teacher-forcing every 10 timesteps. $X \in \mathbb{R}^{n \times n}$ in this case was a directed adjacency matrix over the graph $G = (V, E)$ where $V$ were the nodes. $X_{ij} = 1$ is interpreted as there being an edge from $i \to j$ and 0 for no edge. For the SMTs over undirected edges (*E.F. Ent. Top* $|V| - 1$ and *Spanning Tree*) $X$ was the symmetric, directed adjacency matrix with edges in both directions for each undirected edge. The decoder passed messages between both connected and not-connected nodes. When considering a message from node $i \to j$, it used one network for the edges with $X_{ij} = 1$ and another network for the edges with $X_{ij} = 0$, such that we could differentiate the two edge "types". For the SST relaxation, both messages were passed, weighted by $(X_t)_{ij}$ and $1 - (X_t)_{ij}$, respectively. Because of the parameterization of our model, during evaluation, it is ambiguous whether the sampled hard graph is in the correct representation (adjacency matrix where 1 is the existence of an edge, and 0 is the non-existence of an edge). Therefore, when measuring precision and recall for structure discovery, we selected whichever graph (the sampled graph versus the graph with adjacency matrix of one minus that of the sampled graph) that yielded the highest precision, and reported precision and recall measurements for that graph.

**Objective**   Our ELBO objective consisted of the reconstruction error and KL divergence. The reconstruction error was the Gaussian log likelihood of the predicted node positions generated from the decoder given ground truth node positions. As mentioned in D.4, we computed the KL divergence with respect to $U$ instead of the sampled graph for all methods, because computing the probability of a *Spanning Tree*, or *Top* $k$ sample is not computationally efficient. We chose our prior to be $\text{Gumbel}(0)$. The KL divergence between a Gumbel distribution with location $\theta$ and a Gumbel distribution with location 0, is $\theta + \exp(-\theta) - 1$.

### D.5.3 Training

All graph layout experiments were run with batch size 128 for 50000 steps. We evaluated the model on the validation set every 500 training steps, and saved the model that achieved the best average validation ELBO. We used the Adam optimizer with a constant learning rate, and $\beta_1 = 0.9, \beta_2 = 0.999, \epsilon = 10^{-8}$. We tuned hypermarameters using random uniform search over a hypercube-shaped search space with 20 trials. We tuned the constant learning rate, and temperature $t$ for all methods. For *E.F. Ent. Top* $k$, we additionally tuned $\epsilon$, which is used when computing the gradients for the backward-pass using finite-differences. The ranges for hyperparameter values were chosen such that optimal hyperparameter values (corresponding to the best validation ELBO) were not close to the boundaries of the search space.

## D.6 Unsupervised Parsing on ListOps

### D.6.1 Data

We considered a simplified variant of the ListOps dataset [62]. Specifically, we used the same data generation process as [62] but excluded the `summod` operator and used rejection sampling to ensure that the lengths of all sequences in the dataset ranged only from 10 to 50 and that our dataset contained

the same number of sequences of depths $d \in \{1, 2, 3, 4, 5\}$. Depth was measured with respect to the ground truth parse tree. For each sequence, the ground truth parse tree was defined by directed edges from all operators to their respective operands. We generated 100,000 samples for the training set (20,000 for each depth), and 10,000 for the validation and test set (2,000 for each depth).

### D.6.2 Model

We used an embedding dimension of 60, and all neural networks had 60 hidden units.

**LSTM** We used a single-layered LSTM going from left to right on the input embedding matrix. The LSTM had hidden size 60 and includes dropout with probability 0.1. The output of the LSTM was flattened and fed into a single linear layer to bring the dimension to 60. The output of the linear layer was fed into an MLP with one hidden layer and ReLU activations.

**GNN on latent (di)graph** Our models had two main parts: an LSTM encoder that produced a graph adjacency matrix sample ($X$ or $X_t$), and a GNN that passed messages over the sampled graph.

The LSTM encoder consisted of two LSTMs– one representing the "head" tokens, and the other for "modifier" tokens. Both LSTMs were single-layered, left-to-right, with hidden size 60, and include dropout with probability 0.1. Each LSTM outputted a single real valued vector for each token $i$ of $n$ tokens with dimension 60. To obtain $\theta \in \mathbb{R}^{n \times n}$, we defined $\theta_{ij} = v_i^{\mathrm{head}T} v_j^{\mathrm{mod}}$, where $v_i^{\mathrm{head}}$ is the vector outputted by the head LSTM for word $i$ and $v_i^{\mathrm{mod}}$ is the vector outputted by the modifier LSTM for word $j$. As in the graph layout experiments, we symmetrized $\theta$ for the SSTs that require undirected edges (*Indep. Undirected Edges*, and *Spanning Tree*). For exponential $U \sim \mathrm{Exp}(\theta)$, $\theta$ was parameterized as the softplus function of the $\mathbb{R}^{n \times n}$ matrix output of the encoder. We used the Torch-struct library [73] to obtain soft samples for *arborescence*.

$X \in \mathbb{R}^{n \times n}$ in this case was a directed adjacency matrix over the graph $G = (V, E)$ where $V$ were the tokens. $X_{ij} = 1$ is interpreted as there being an edge from $i \rightarrow j$ and 0 for no edge. For the SMTs over undirected edges (*Indep. Undirected Edges* and *Spanning Tree*) $X$ was the symmetric, directed adjacency matrix with edges in both directions for each undirected edge. For *Arborescence*, we assumed the first token is the root node of the arborescence.

Given $X$, the GNN ran 5 message passing steps over the adjacency matrix, with the initial node embeddings being the input embedding. The GNN architecture was identical to the GNN decoder in the graph layout experiments, except we did not pass messages on edges with $X_{ij} = 0$ and we did not include the last MLP after every messaging step. For the SST, we simply weighted each message from $i \rightarrow j$ by $(X_t)_{ij}$. We used dropout with probability 0.1 in the MLPs. We used a recurrent connection after every message passing step. The LSTM encoder and the GNN each had their own embedding lookup table for the input. We fed the node embedding of the first token to an MLP with one hidden layer and ReLU activations.

### D.6.3 Training

All ListOps experiments were run with batch size 100 for 50 epochs. We evaluated the model on the validation set every 800 training steps, and saved the model that achieved the best average validation task accuracy. We used the AdamW optimizer with a constant learning rate, and $\beta_1 = 0.9, \beta_2 = 0.999, \epsilon = 10^{-8}$. We tuned hypermarameters using random uniform search over a hypercube-shaped search space with 20 trials. We tuned the constant learning rate, temperature $t$, and weight decay for all methods. The ranges for hyperparameter values were chosen such that optimal hyperparameter values (corresponding to the best validation accuracy) were not close to the boundaries of the search space.

### D.7 Learning To Explain (L2X) Aspect Ratings

**Data.** We used the BeerAdvocate dataset [57], which contains reviews comprised of free-text feedback and ratings for multiple aspects, including appearance, aroma, palate, and taste. For each aspect, we used the same de-correlated subsets of the original dataset as [49]. The training set for the aspect appearance contained 80k reviews and for all other aspects 70k reviews. Unfortunately, [49] do not provide separate validation and test sets. Therefore for each aspect, we split their heldout set into two evenly sized validation and test sets containing 5k reviews each. We used pre-trained word

embeddings of dimension 200 from [49] to initialize all models. Each review was padded/ cut to 350 words. For all aspects, subset precision was measured on the same subset of 993 annotated reviews from [57]. The aspect ratings were normalized to the unit interval $[0, 1]$ and MSE is reported on the normalized scale.

**Model.** Our model used convolutional neural networks to parameterize both the subset distribution and to make a prediction from the masked embeddings. For parameterizing the masks, we considered a simple and (a more) complex architecture. The simple architecture consisted of a Dropout layer (with $p = 0.1$) and a convolutional layer (with one filter and a kernel size of one) to parameterize $\theta_i \in \mathbb{R}$ for each word $i$, producing a the vector $\theta \in \mathbb{R}^n$. For *Corr. Top k*, $\theta \in \mathbb{R}^{2n-1}$. The first $n$ dimensions correspond to each word $i$ and are parameterized as above. For dimensions $i \in \{n + 1, \ldots, 2n - 1\}$, $\theta_i$ represents a coupling between words $i$ and $i + 1$, so we denote this $\theta_{i,i+1}$. It was parameterized as the sum of three terms, $\theta_{i,i+1} = \phi_i + \phi_i' + \phi_{i,i+1}$: $\phi_i \in \mathbb{R}$ computed using a seperate convolutional layer of the same kind as described above, $\phi_i' \in \mathbb{R}$ computed using yet another convolutional layer of the same kind as described above, and $\phi_{i,i+1} \in \mathbb{R}$ obtained from a third convolutional layer with one filter and a kernel size of two. In total, we used four separate convolutional layers to parameterize the simple encoder. For the complex architecture, we used two additional convolutional layers, each with 100 filters, kernels of size three, ReLU activations to compute the initial word embeddings. This was padded to maintain the length of a review.

$X$ was a $k$-hot binary vector in $n$-dimensions with each dimension corresponding to a word. For *Corr. Top K*, we ignored dimensions $i \in \{n + 1, \ldots, 2n - 1\}$, which correspond to the pairwise indicators. Predictions were made from the masked embeddings, using $X_i$ to mask the embedding of word $i$. Our model applied a soft (at training) or hard (at evaluation) subset mask to the word embeddings of a given review. Our model then used two convolutional layers over these masked embeddings, each with 100 filters, kernels of size three and ReLU activations. The resulting output was max-pooled over all feature vectors. Our model then made predictions using a Dropout layer (with $p = 0.1$), a fully connected layer (with output dimension 100, ReLU activation) and a fully connected layer (with output dimension one, sigmoid activation) to predict the rating of a given aspect.

**Training.** We trained all models for ten epochs at minibatches of size 100. We used the Adam optimizer [36] and a linear learning rate schedule. Hyperparameters included the initial learning rate, its final decay factor, the number of epochs over which to decay the learning rate, weight decay and the temperature of the relaxed gradient estimator. Hyperparameters were optimized for each model using random search over 25 independent runs. The learning rate and its associated hyperparameters were sampled from $\{1, 3, 5, 10, 30, 50, 100\} \times 10^{-4}$, $\{1, 10, 100, 1000\} \times 10^{-4}$ and $\{5, 6, \ldots, 10\}$ respectively. Weight decay was sampled from $\{0, 1, 10, 100\} \times 10^{-6}$ and the temperature was sampled from $[0.1, 2]$. For a given run, models were evaluated on the validation set at the end of each epoch and the best validated model was was retained. For *E.F. Ent. Top k* and *Corr. Top k*, we trained these methods with $\epsilon \in \{1, 10, 100, 1000\} \times 10^{-3}$ and selected the best $\epsilon$ on the validation set. We believe that it may be possible to improve on the results we report with an efficient exact implementation of the backward pass for these two methods. We found the overhead created by automatic differentiation software to differentiate through the unrolled dynamic program was prohibitively large in this experiment.