[Reviews · NeurIPS 2020]

Review 1

Summary and Contributions: The paper proposes a unified framework for relaxations of samples from discrete distributions, which makes them amenable to gradient learning. The main idea is to combine perturbation method for sampling from discrete distributions continuous relaxation of corresponding linear optimisation problem. Authors propose several continuous relaxations and apply them to combinatorial spaces like graph spanning trees. They show experimentally that proposed methods perform better then existing ones on structure discovery, i.e. recovering latent structure with Variational inference learning.

Strengths: Proposed method is a generalization of existing relaxations of discrete distributions. The approach unifies existing approaches and gives a systematic way of analysing them as well as for finding new ones. All the claims are supported by extensive theoretical derivations and empirical evidence. Given the importance of learning structured hidden representations the work is a significant advancement in this direction.

Weaknesses: The proposed method is limited by the ability to solve corresponding relaxed optimization problem, however it still gives access to a large number of combinatorial spaces with current optimization methods

Correctness: The claims are sound and correct, accompanied by solid empirical evidence

Clarity: The paper is well organized and clear

Relation to Prior Work: Authors are very careful in differentiating their contribution from prior work and acknowledging it

Reproducibility: Yes

Additional Feedback: ******post rebutal********** I'd like to thank the authors for their response and I will keep my score unchanged.


Review 2

Summary and Contributions: The paper proposes a framework to construct relaxations for structured discrete random variables. The framework both unifies several previously proposed relaxations and builds alternatives for bespoke structured relaxations. In the experimental section, the authors train several latent variable models and show that the models with structured variables outperform less structured baselines.

Strengths: The authors generalize the approach used to construct the Gumbel-Sinkhorn relaxations and give a recipe for creating novel relaxations for other domains. There are quite a few bespoke relaxations for structured variables in the machine learning literature, suggesting that the topic is valuable and relevant to the NeurIPS community. Using the recipe, the authors "fill the gaps" by constructing several novel relaxations, showing that the general scheme is applicable.

Weaknesses: Typically, one expects a reparametrizable distribution with a tractable density for training with the evidence lower bound as an objective. For example, the concrete distribution is reparametrizable and has a tractable density. While the SST allows reparametrization, the authors do not discuss it's density in the paper. In the experiments, to compute the evidence lower bound, the authors resort to the density of the random variable used for the SST, but not the density SST output (i.e., density of U in eq. 6 instead of density of X_t). How justifiable is this substitute? Finally, for some reason, the paper does not compare against the Perturb & Parse algorithm.

Correctness: I did not find any incorrect claims or any methodological issues in the paper.

Clarity: So the paper is well written, densely packed with valuable references and was pleasant to read.

Relation to Prior Work: The authors discuss how their work differs from previous contributions. In particular, they propose a general framework, while previous works considered generalizations of the Gumbel-Softmax trick for narrow specific domains.

Reproducibility: Yes

Additional Feedback: [Post rebuttal edit: Thank you for comments. I am keeping my original score after the rebuttal.] The phrase starting on line 62 seems needlessly confusing. Although argmax is, in fact, differentiable almost everywhere, it is not differentiable in the sense of being differentiable at every point. The Leibniz rule requires the existence of partial derivative for all parameter values. So the lack of differentiability seems to be a problem in the sense of invalidating the Leibniz rule. Do the authors refer to a more general scheme (say, using the Leibniz rule separately for several regions) that is being invalidated by the jump discontinuities(but not the non-differentiability)?


Review 3

Summary and Contributions: Update after the author response: I want to thank the authors for clarifying how exactly KL between prior and approximate posterior is calculated in VI set-up. Usually, an "interesting" inductive bias / prior distribution is formulated in the original combinatorial space X rather than utility space U. Hence, I believe it would be beneficial for the potential reader if this limitation is mentioned explicitly in the paper. ================================================ The paper is concerned with the task of estimating the gradient of the following form: d E_{X~p_\theta}[L(X)] / d\theta. Where X represents a combinatorial object (e.g. one-hot vectors, subsets, spanning trees, etc.). This loss is ubiquitous in variational inference approach to latent variable models with structured latent variables. Also, such loss can be part of the model that encodes specific inductive biases using structured variables. To solve this task, the authors proposed to frame sampling X~p_\theta as solving an integer linear program with random utility weights U_\theta having previously decided on an embedding/encoding of the combinatorial object. This process is called stochastic argmax trick(SMT). The usage of stochastic softmax trick (SST) is proposed to remove the discontinuous behaviour of the argmax. An SST relaxes an SMT by expanding state space of ILP to a convex polytope and adding a strongly convex regularizer. These steps make the solution of LP a continuous/differentiable function of random utility weights, which allows biased estimation of the original gradient with hopefully low variance. The authors confirm the validity of the proposed approach using several different tasks. Namely, Neural Relational Inference where they build a generative model with a latent variable being a graph/spanning-tree. Simplified ListOps task where a graph/spanning-tree/arborescence represents the latent component of the model, and "Learning To Explain"-type of task where the latent component of a classifier/regressor was a k-subset selection.

Strengths: Having the ability to train models with structured latent variables is crucial for introducing specific inductive biases into the model and/or improving its interpretability. This paper makes an important step towards this goal. The proposed method is relatively easy to implement using present automatic differentiation frameworks. The authors provide useful "cookbook" on how to apply SMT and SST for a family of interesting combinatorial objects.

Weaknesses: As the authors noted, it is not straightforward to design SMT trick for a given distribution p_\theta. Instead, they are turning the very problem on its head and treat reparametrization as a primary object and induced/associated implicit p_\theta as the secondary one. This is not a weakness of the approach per se, but it definitely limits the usage of the proposed approach in the variational setup where usually KL between posterior and prior has to be estimated/evaluated and thus the exact/explicit form of p_\theta is needed. In section 6, the authors claim, that the novelty of their work is neither the perturbation model framework nor the relaxation framework in isolation but their combined use for gradient estimation. However, I believe that the same principle has been already used in prior work (e.g. [1] where potentials of the tree are perturbed with noise and relaxed dynamic programming is used as a softmax). Nevertheless, I think that the community will still benefit from the current submission as it provides a very rigorous exposition of the problem and provides some novel relaxations. [1] Learning Latent Trees with Stochastic Perturbations and Differentiable Dynamic Programming. Caio Corro and Ivan Titov, ACL2019

Correctness: The proposed method appears to be sound and correct as well as empirical methodology.

Clarity: The paper is well written and easy to follow.

Relation to Prior Work: The prior work is extensively discussed throughout the paper.

Reproducibility: Yes

Additional Feedback: As far as I understood p_\theta distribution is implicit, so I am puzzled about how KL-part of ELBO for spanning tree is calculated in Table 1.

[Author Response · NeurIPS 2020]

We thank the reviewers for their time and thoughtful feedback. We were glad to see that the reviewers had uniformly positive things to say about the work. Below, we clarify two specific questions.

1. **Computing KL divergences.** R3 and R4 raised concerns about computing the KL in the absence of a tractable probability density of $X_t$ (induced by the choice of random utility $U$).

   R4: "As far as I understood $p_\theta$ distribution is implicit, so I am puzzled about how KL-part of ELBO for spanning tree is calculated in Table 1."

   We computed the KL with respect to the random utility $U$. There are at least 3 possible KL terms for a relaxed ELBO: the KL from a prior over $\mathcal{X}$ to the distribution of $X$, the KL from a prior over $\operatorname{conv}(\mathcal{X})$ to the distribution of $X_t$, or the KL from a prior over $\mathbb{R}^n$ to the distribution of $U$, see Section C.3 of [52, Maddison et al.] for a discussion of this topic. In our case, since we do not know of an explicit tractable density of $X_t$ or $X$, we compute the KL with respect to $U$. This means that for the graph layout experiments, in both training and testing, we computed the KL between $U \sim \operatorname{Gumbel}(\theta)$ and the Gumbel with location 0 prior (see section D.4.2).

   R3: "How justifiable is this substitute?"

   The KL divergence with respect to $U$ is an *upper-bound* to the KL divergence with respect to $X_t$, due to a data processing inequality; therefore, the ELBO that we are optimizing is a *lower-bound* to the relaxed variational objective. Whether or not this is a good choice is an empirical question, but it seems to not be an issue for the graph layout experiments. Note, that *when optimizing the relaxed objective*, using a KL divergence with respect to $X$ does not result in a lower-bound to the relaxed variational objective, as it is not necessarily an ELBO for the continuous relaxed model (see again Section C.3 of [52]).

2. **Differentiability of the argmax.**

   R3: "Do the authors refer to a more general scheme that is being invalidated by the jump discontinuities (but not the non-differentiability)?"

   That's a great question. We agree that we should make this clearer. Indeed, we refer to Proposition 2.3 in Chapter 7.2 in [8, Asmussen & Glynn]. This gives a slightly more general condition for the exchange of expectation and differentiation than Leibniz rule. It does not require the existence of partial derivatives everywhere, but instead poses a Lipschitz condition. The jump discontinuities in SMTs violate this condition, see also Remark 2.6 in [8]. In contrast, the Euclidean projection for SSTs does not. As a result, it admits a reparameterization gradient, even though its partial derivatives do not exist everywhere.

If accepted, we plan to use the additional page allowance in the following way. First, we would add text addressing the clarifications requested by the reviewers. Second, we've since explored additional score function estimator baselines. We would add these to the appendix of the graph layout experiments. The baselines improved with our efforts, but the conclusions are unchanged (SSTs outperforming the other gradient estimator baselines). Finally, we would use the additional page allowance to add Fig. 1 (below) to aid in intuition.

Figure 1: Structured discrete objects can be represented by binary arrays. In these graphical representations, color indicates 1 and no color indicates 0. For example, "Spanning tree" is the adjacency matrix of an undirected spanning tree over 6 nodes; "Arborescence" is the adjacency matrix of a directed spanning tree rooted at node 3.

[Meta-Review · NeurIPS 2020]

All three referees are very positive about this paper and support accept. Please follow Reviewer 4's additional comments to clarify how the KL term is calculated in the revision.